# Mem-T: Densifying Rewards for Long-Horizon Memory Agents

**Yanwei Yue** [* 1 2]  **Guibin Zhang** [*]  **Boci Peng** [1 2]  **Xuanbo Fan** [1 2]  **Jiaxin Guo** [1 2]  **Qiankun Li** [3 4]  **Yan Zhang** [1 2]

## Abstract

Memory agents, which depart from predefined memory-processing pipelines by endogenously managing the processing, storage, and retrieval of memories, have garnered increasing attention for their autonomy and adaptability. However, existing training paradigms remain constrained: agents often traverse long-horizon sequences of memory operations before receiving sparse and delayed rewards, which hinders truly end-to-end optimization of memory management policies. To address this limitation, we introduce Mem-T, an autonomous memory agent that interfaces with a lightweight hierarchical memory database to perform dynamic updates and multi-turn retrieval over streaming inputs. To effectively train long-horizon memory management capabilities, we further propose MoT-GRPO, a tree-guided reinforcement learning framework that transforms sparse terminal feedback into dense, step-wise supervision via memory operation tree backpropagation and hindsight credit assignment, thereby enabling the joint optimization of memory construction and retrieval. Extensive experiments demonstrate that Mem-T is **(1) high-performing**, surpassing frameworks such as A-Mem and Mem0 by up to $14.92\%$, and **(2) economical**, operating on a favorable accuracy-efficiency Pareto frontier and reducing inference tokens per query by $\sim 24.45\%$ relative to GAM without sacrificing performance. Codes are available at https://github.com/yanweiyue/Mem-T.

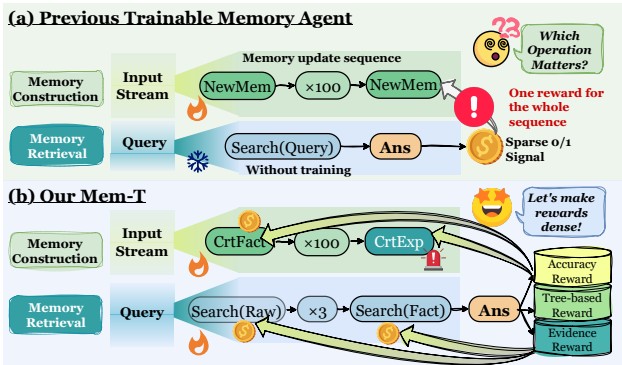

*Figure 1.* The paradigm comparison between the previous trainable memory agent and Mem-T.

## 1. Introduction

As Large Language Models (LLMs) rapidly evolve into powerful AI agents, they have achieved significant success across various fields (Hong et al., 2024; Wu et al., 2023; Qian et al., 2025; Yang et al., 2025; Xu & Peng, 2025). However, constrained by the finite context windows of foundation models, AI agents face inherent challenges with long-term inconsistency (Li et al., 2024; Liu et al., 2025) and context forgetting during extended multi-turn interactions (Ai et al., 2025; Liu et al., 2025). As a promising frontier, memory systems dynamically construct and leverage memories from historical interactions (Li et al., 2025b; Fang et al., 2025), thereby sustaining temporal coherence and long-term intelligence beyond finite context windows (Ye et al., 2025b; Zhao et al., 2024), and have consequently emerged as a core component of modern agentic systems (Chhikara et al., 2025; Zhang et al., 2025).

Tracing the evolution of memory systems, early frameworks such as MemGPT (Packer et al., 2023), Mem0 (Chhikara et al., 2025), and A-Mem (Xu et al., 2025) predominantly rely on hand-crafted prompts and heuristic rules to guide frozen LLMs in populating predefined memory structures. As a result, their performance is inherently bounded by the base model's instruction-following capacity and rigid human priors, often leading to suboptimal outcomes (Xiong et al., 2025; Wu et al., 2025a). By contrast, recent approaches such as Memory-R1 (Yan et al., 2025b), Mem-$\alpha$ (Wang et al., 2025), and MemTool (Lumer et al., 2025) employ reinforcement learning (e.g., GRPO (Shao

---

[*]Equal contribution  [1]School of Intelligence Science and Technology, Peking University  [2]State Key Laboratory of General Artificial Intelligence, Peking University, Beijing, China  [3]Nanyang Technological University  [4]IGS, Imperial College London. Correspondence to: Qiankun Li <q.li2@imperial.ac.uk>, Yan Zhang <zhyzhy001@pku.edu.cn>, Main Contact: Yanwei Yue <ywyue25@stu.pku.edu.cn>.

*Proceedings of the 43rd International Conference on Machine Learning*, Seoul, South Korea. PMLR 306, 2026. Copyright 2026 by the author(s).

et al., 2024)) to train LLMs into adaptive policies for dynamic memory curation and retrieval, commonly referred to as memory agents. This shift constitutes a fundamental paradigm change, recasting memory management from static instruction adherence into a problem of adaptive policy optimization (Hu et al., 2026b).

However, current paradigms for training memory agents remain fundamentally constrained by temporal credit assignment (Pignatelli et al., 2024), *i.e.*, the challenge of attributing sparse and delayed rewards to causative actions along long-horizon memory operation sequences. This limitation is particularly acute in memory-centric tasks, where agents may execute hundreds of memory operations across ~500 turns within million-token contexts before receiving a binary $0/1$ reward derived from sporadic QA accuracy signals (Wu et al., 2025a; Tan et al., 2025). Existing approaches fail to bridge this gap, as they indiscriminately propagate the sparse terminal reward across all memory operations without dense supervision or process-level attribution (Yan et al., 2025b; Wang et al., 2025). Consequently, this extreme sparsity impedes effective optimization of the full memory operation trajectory. To put it more formally:

> *How can we implement a fully trainable memory agent framework that jointly optimizes memory construction and retrieval, supervised with dense rewards and accurate process-level attribution?*

To address this challenge, we introduce `Mem-T`, a streamlined hierarchical memory agent optimized under a process-supervised, attribution-centric training paradigm termed **Memory Operation Tree-guided GRPO** (MoT-GRPO). Functionally, `Mem-T` integrates three core capabilities: (i) formation and (ii) evolution operations that maintain and refine the hierarchical memory database over dynamic information streams, and (iii) a retrieval operation that conducts multi-turn, autonomous search to provide accurate memory clues. To jointly optimize these components, MoT-GRPO employs a dual-track training mechanism integrating memory retrieval and construction. To refine memory retrieval, it constructs multiple Memory operation Trees (MoT) to explore diverse trajectories, leveraging the branching topology to back-propagate sparse outcome rewards to intermediate nodes, thereby generating dense process-level signals and identifying critical search paths. To refine memory construction, the utility of the MoT is explicitly attributed back to source memory items via hindsight credit assignment, supervising the corresponding formation and evolution operations. This paradigm effectively mitigates reward sparsity and attribution ambiguity, rendering memory interactions both interpretable and learnable. Our contributions can be summarized as:

- *Unified Memory Framework.* We propose `Mem-T`, a streamlined memory management agent with a hierarchi-

cal architecture that integrates factual, experiential, and working memory, and agentically orchestrates the full lifecycle of memory operations.

- *Joint Memory Optimization.* We present MoT-GRPO, a learning paradigm that enables the joint optimization of memory construction and retrieval. It introduces tree-based reward densification and credit assignment into memory retrieval training. Furthermore, it performs credit assignment on the corresponding memory construction operations, using the inference rewards associated with specific memory items as pivots.

- *Experimental Evaluation.* Comprehensive evaluations on four memory benchmarks demonstrate that `Mem-T` achieves state-of-the-art performance while maintaining a superior Pareto frontier, delivering up to $14.92\%$ F1 gains and reducing inference tokens per query by $\sim 24.45\%$ compared with GAM and A-Mem baselines.

## 2. Related Work

*Table 1.* Comparison of different memory agent systems. ✖: Not included; ✂: Included but heuristic-based; 🔥: Included and trainable. **Abbreviations**: Fact.=Factual Memory, Exp.=Experiential Memory, Work.=Working Memory, Form.=Memory Formation, Evol.=Memory Evolution, Retr.=Memory Retrieval, Proc. Attr.=Process Attribution.

| Method | Functions | | | Operations | | | Proc. Attr. |
|---|---|---|---|---|---|---|---|
| | Fact. | Exp. | Work. | Form. | Evol. | Retr. | |
| MemAgent | ✖ | ✖ | 🔥 | 🔥 | 🔥 | ✖ | ✖ |
| Context-Folding | ✖ | ✖ | 🔥 | 🔥 | ✂ | ✖ | ✖ |
| Memory-R1 | 🔥 | ✖ | ✖ | ✂ | 🔥 | ✂ | ✖ |
| Mem-$\alpha$ | 🔥 | ✖ | 🔥 | 🔥 | ✂ | ✖ | ✖ |
| MemSearcher | 🔥 | ✖ | ✖ | ✖ | ✖ | 🔥 | ✖ |
| LightSearcher | ✖ | 🔥 | ✖ | ✂ | ✖ | 🔥 | ✖ |
| `Mem-T` | 🔥 | 🔥 | 🔥 | 🔥 | 🔥 | 🔥 | 🔥 |

**Memory Agent Architectures.** In recent years, memory agents have advanced rapidly, evolving from heuristic-based systems such as MemoryBank (Zhong et al., 2024) and MemGPT (Packer et al., 2023) to more agentic architectures, including Mem0 (Chhikara et al., 2025), MemOS (Li et al., 2025b), and A-Mem (Xu et al., 2025). **Functionally**, prior work spans three categories: *(I) Factual Memory*, preserving declarative knowledge for long-term consistency (Zhong et al., 2024); *(II) Experiential Memory*, distilling experience from trajectories to support continual self-improvement (Zhao et al., 2024); and *(III) Working Memory*, managing dynamic context for ongoing tasks (Wu et al., 2025b). **Operationally**, the memory lifecycle comprises *(I) Formation*, transforming raw context into high-value memory; *(II) Evolution*, integrating new insights with existing memory store; and *(III) Retrieval*, performing accurate retrieval from the memory base. As shown in Table 1,

our `Mem-T`, despite its streamlined design, spans all three functional classes and all three operational stages.

**Reinforcement Learning for Memory Agents.** As memory systems scale in complexity, the efficacy of foundation models in managing memory increasingly becomes the primary performance bottleneck. Consequently, reinforcement learning (RL) has emerged as a central paradigm for endowing LLMs with adaptive memory management capabilities (Hu et al., 2026b). Current research spans a broad spectrum, from short-term working memory to long-term factual and experiential memory. *Working Memory.* RL has been used to enable agents to autonomously manage execution context within a single task (Yu et al., 2025; Chen et al., 2025a), particularly in settings such as deep research and web browsing (Zhou et al., 2025; Sun et al., 2025; Ye et al., 2025a). *Long-term Factual Memory.* Prior work targets different stages of memory management: Memory-R1 (Yan et al., 2025b) emphasizes memory evolution, Mem-$\alpha$ (Wang et al., 2025) addresses both formation and evolution, and MemSearcher (Yuan et al., 2025) focuses on training agents to exploit retrieval tools. *Long-term Experiential Memory.* Methods such as LightSearcher (Lan et al., 2025) and MemRL (Zhang et al., 2026) improve the acquisition, refinement, and reuse of skills over time. Despite these advances, RL-based approaches remain limited by sparse rewards and temporal credit assignment in long-horizon settings, hindering effective optimization across the full memory construction and utilization pipeline, as shown in Table 1.

**Tree-Based RL.** Tree-based RL organizes the reasoning process into a tree structure (Zhang et al., 2024; Hou et al., 2025). By leveraging the step-level signals naturally generated from the tree structure to achieve process supervision, it simultaneously addresses the challenges of sparse reward and inefficient exploration (Li et al., 2025a). However, tree-attributed RL (*e.g.*, Tree-GRPO (Ji et al., 2025)) faces difficulties when adapted to memory agent training; its fully online tree-structured sampling is computationally prohibitive for long-horizon memory operations. Moreover, it fails to resolve the extremely sparse dependency between the outcome and the few relevant early memory constructions. To address these challenges, MoT-GRPO adopts traditional tree sampling during the short-horizon memory retrieval phase; however, for the long-horizon memory construction phase, we employ memory-pivot and evidence-guided hindsight credit assignment.

## 3. Method

### 3.1. Mem-T Workflow

**Hierarchical Memory Definition.** We consider the agent interacting with a continuous information stream $\mathcal{X} = \{x_1, x_2, \ldots, x_T\}$. At each time step $t$, corresponding to the processing of the current chunk $x_t$, the system maintains a hierarchical memory state $\mathcal{M}_t$:

$$\mathcal{M}_t = \langle \mathcal{M}_t^{\text{work}}, \mathcal{M}_t^{\text{fact}}, \mathcal{M}_t^{\text{exp}}, \mathcal{M}_t^{\text{raw}} \rangle. \quad (1)$$

Within this hierarchy, **Working Memory** ($\mathcal{M}_t^{\text{work}}$) iteratively updates a concise summary at each step, maintaining within-episode coherence. The long-term memory consists of three modules: **Factual Memory** ($\mathcal{M}_t^{\text{fact}}$) stores declarative knowledge, **Experiential Memory** ($\mathcal{M}_t^{\text{exp}}$) captures procedural knowledge, and **Raw Memory** ($\mathcal{M}_t^{\text{raw}}$) archives raw data across sessions. Formally, we have:

$$\mathcal{M}_t^{\text{fact}} = \{m_i^{\text{fact}} \mid m_i^{\text{fact}} = (f_i, t_i^{\text{start}}, t_i^{\text{end}})\}_{i=1}^{N_f},$$
$$\mathcal{M}_t^{\text{exp}} = \{m_j^{\text{exp}} \mid m_j^{\text{exp}} = (e_j, t_j^{\text{start}}, t_j^{\text{end}})\}_{j=1}^{N_e}, \quad (2)$$
$$\mathcal{M}_t^{\text{raw}} = \{m_l^{\text{raw}} \mid m_l^{\text{raw}} = (x_l, t_l^{\text{raw}})\}_{l=1}^{t},$$

where each $m^{(\cdot)}$ represents an atomic memory unit. Specifically, $f_i$ and $e_j$ represent concrete facts and strategies, respectively, bound by validity time windows $[t^{\text{start}}, t^{\text{end}}]$.

**Memory Operation Pipeline.** Building upon this hierarchical memory, we formulate the agent's interaction as a dual-track decision process, comprising continuous memory construction and on-demand memory utilization.

**Phase I: Continuous Memory Construction.** As the agent processes the input stream $x_t$, it proactively constructs new memory candidates via the memory formation policy $\pi_{\text{form}}$. This policy scans the raw input to identify salient information and operates on the formation action space $\mathcal{A}_{\text{form}} = \{\texttt{CrtFact}, \texttt{CrtExp}, \texttt{CrtRaw}, \texttt{UpdWork}\}$. Here, `CrtFact`, `CrtExp`, and `CrtRaw` create atomic declarative facts, procedural strategies, and raw data, respectively, while `UpdWork` updates the session-level working summary. Formally, the formation process is defined as:

$$a_{\text{form}} \sim \pi_{\text{form}}(\cdot | x_t, \mathcal{M}_t^{\text{work}}), \quad a_{\text{form}} \subseteq \mathcal{A}_{\text{form}},$$
$$\mathcal{M}_t^{\text{cand}} = \{m \mid m \leftarrow \text{Execute}(op), \forall op \in a_{\text{form}}\}, \quad (3)$$

where $\mathcal{M}_t^{\text{cand}}$ denotes the set of candidate memories extracted from $x_t$. For each candidate $m \in \mathcal{M}_t^{\text{cand}}$, the memory evolution policy $\pi_{\text{evol}}$ integrates it into $\mathcal{M}_t$. Specifically, the policy considers memories in $\mathcal{M}_t$ that are relevant to $m$, and samples an evolution action $a_{\text{evol}} \sim \pi_{\text{evol}}(\cdot \mid m, \mathcal{M}_t)$ from the action space $\mathcal{A}_{\text{evol}} = \{\texttt{ADD}, \texttt{UPDATE}, \texttt{DELETE}, \texttt{IGNORE}\}$. Collectively, these actions define the set of memories to be added ($\Delta^+$) and removed ($\Delta^-$) from the memory store:

$$\Delta^+ = \{m | a_{\text{evol}} = \texttt{ADD}\} \cup \{m_{\text{refined}} | a_{\text{evol}} = \texttt{UPDATE}\},$$
$$\Delta^- = \{m_{\text{target}} | a_{\text{evol}} = \texttt{DELETE}\} \cup \{m_{\text{old}} | a_{\text{evol}} = \texttt{UPDATE}\}. \quad (4)$$

Consequently, the memory store is updated accordingly:

$$\mathcal{M}_{t+1} = (\mathcal{M}_t \setminus \Delta^-) \cup \Delta^+. \quad (5)$$

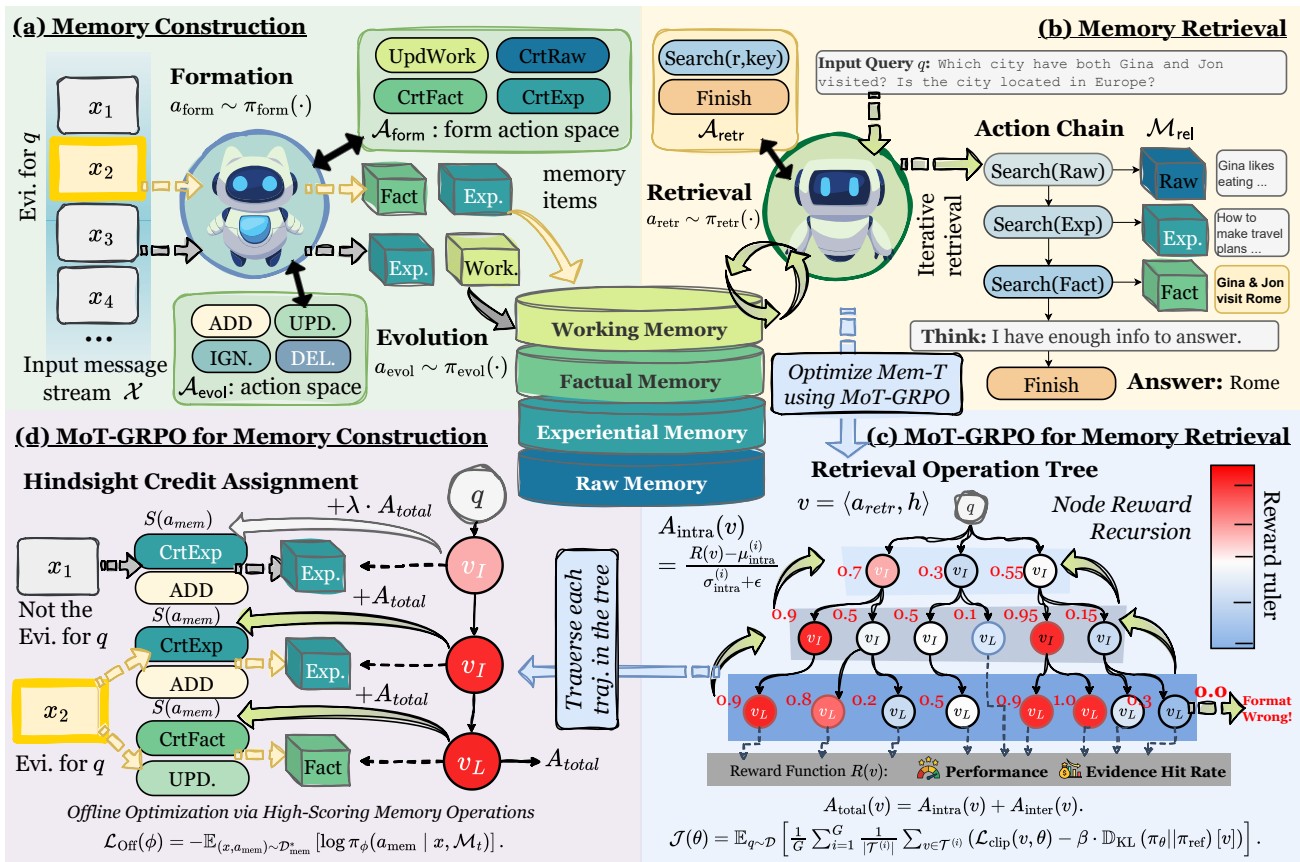

*Figure 2.* The overall framework of our proposed `Mem-T`.

**Phase II: On-Demand Memory Retrieval.** Based on the constructed memory store $\mathcal{M}_t$, when a query $q$ arises, the agent employs a multi-turn retrieval to respond. During this process, the memory retrieval policy $\pi_{\text{retr}}$ selects actions from the retrieval action space $\mathcal{A}_{\text{retr}}$, which includes queries for each memory module and a terminal signal:

$$\mathcal{A}_{\text{retr}} = \{\texttt{Search}(r, \text{key}, \text{topk}) \mid r \in \mathcal{M}_t\} \cup \{\texttt{Finish}\},$$
(6)

where $r$ is the memory type to be retrieved, key is the retrieval query. Unlike single-step retrieval, $\pi_{\text{retr}}$ operates as a sequential decision policy. At each step $k$, conditioned on the query $q$ and the history context $h_{k-1}$, the policy samples an action $a_k$:

$$a_k \sim \pi_{\text{retr}}(\cdot \mid q, \mathcal{M}_t, h_{k-1}), \quad a_k \in \mathcal{A}_{\text{retr}}. \quad (7)$$

Here, the history context $h_{k-1}$ consists of the retrieved relevant memory set $\mathcal{M}_{k-1}^{\text{rel}}$ and the model-generated reasoning traces $z_{k-1}$. This iterative process accumulates the relevant memory set $\mathcal{M}^{\text{rel}}$ by aggregating the observations from each search step. Finally, the loop terminates when the policy selects the `Finish` action, signaling that the gathered information is sufficient to support the final answer $y \sim P_\theta(\cdot \mid q, \mathcal{M}^{\text{rel}})$.

Through this holistic framework, `Mem-T` delivers the first

unified memory framework spanning all four functional memory types (Factual, Experiential, Working, Raw Memory) and the full memory lifecycle (Formation, Evolution, Retrieval), making it highly adaptable for long-horizon dialogues and long-horizon agent tasks.

### 3.2. MoT-GRPO for Memory Retrieval

In long-horizon scenarios, memory operation chains become extremely long, making credit assignment and reward sparsity major challenges. To address these issues, we propose Memory Operation Tree GRPO (MOT-GRPO). For memory retrieval training, we pioneer the application of tree-based RL paradigms (Shao et al., 2024; Ji et al., 2025) to optimize memory retrieval operations.

**Memory Operation Tree Construction.** In the retrieval phase, to achieve efficient rollout generation while obtaining dense intermediate signals, we employ an **Iterative Branching Rollout** to construct the Memory Operation Tree(MoT). Formally, we define a node in MoT as a tuple $v = \langle a_{retr}, h \rangle$, representing a specific operation $a_{retr} \in \mathcal{A}_{\text{retr}}$ and the reasoning context $h$.

For each query, we initialize an ensemble of $G$ independent

MoTs $\{\mathcal{T}_0^{(i)}\}_{i=1}^G$. Each tree $\mathcal{T}_0^{(i)}$ initially contains a single seed trajectory $\tau^{(i)}$, obtained by a full rollout from the root state $(q, \mathcal{M}_t, h_0 = \emptyset)$:

$$
\begin{aligned}
\mathcal{T}_0^{(i)} &= \{\tau^{(i)}\}, \quad i = 1, \ldots, G \\
\tau^{(i)} &= (v_1^{(i)}, v_2^{(i)}, \ldots, v_{L_i}^{(i)}), \text{where } v_k^{(i)} = \langle a_k^{(i)}, h_k^{(i)} \rangle.
\end{aligned}
\tag{8}
$$

Subsequently, we iteratively densify each $\mathcal{T}^{(i)}$ over $M$ expansion rounds. In each expansion round $j \in \{1, \ldots, M\}$, we stochastically sample $N_v$ non-terminal pivot nodes $\{v_n^*\}_{n=1}^{N_v}$ from each tree $\mathcal{T}_{j-1}^{(i)}$. For each node $v^*$ and its corresponding context history $h_{v^*}$, the policy executes a new rollout to generate a branch trajectory $\tau_{\text{branch}}$:

$$
\begin{aligned}
\tau_{\text{branch}} &\sim \pi_{\text{retr}}(\cdot \mid q, \mathcal{M}_t, h_{v^*}), \\
\tau_{\text{new}} &= \text{Path}(v^*) \oplus \tau_{\text{branch}}.
\end{aligned}
\tag{9}
$$

The newly generated trajectories are then grafted onto the tree, updating its state to $\mathcal{T}_j^{(i)}$. After $M$ rounds, this process yields a final ensemble of $G$ MoTs $\{\mathcal{T}_M^{(i)}\}_{i=1}^G$.

**Node-wise Reward Backpropagation.** Instead of relying solely on sparse terminal rewards, we assign a dense reward $R(v)$ to every node $v$, synthesizing immediate retrieval quality with expected future success. Formally, for a node $v$ with retrieved memories $\mathcal{M}_v^{\text{rel}}$, we define the reward as:

$$
R(v) = \mathbb{I}_{\text{fmt}}(v) \cdot (\alpha \cdot \text{Evid}(v) + \text{Perform}(v))
\tag{10}
$$

Here, $\mathbb{I}_{\text{fmt}}(v)$ serves as a binary validity mask ensuring syntactic correctness of tool invocations; $\text{Evid}(v)$ measures the immediate evidence density, calculated as the proportion of ground-truth evidence retrieved in $\mathcal{M}_v^{\text{rel}}$; and $\text{Perform}(v)$ denotes the expected terminal performance of node $v$. For a leaf node, it is defined as the answer quality measured by the F1 score or accuracy. For an internal node, it is computed as the average $\text{Perform}(\cdot)$ over all its child nodes $\text{Ch}(v)$:

$$
\text{Perform}(v) = \begin{cases} \text{F1}(v), & v \in \mathcal{V}_{\text{leaf}}, \\ \frac{1}{|\text{Ch}(v)|} \sum_{u \in \text{Ch}(v)} \text{Perform}(u), & \text{otherwise}. \end{cases}
$$

This formulation ensures that high-reward nodes should adhere to valid formats, retrieve relevant evidence, and lead to high-quality outcomes.

**Dual-Scale Advantage Estimation.** To enable tree-based credit assignment, we perform grouped advantage estimation at both the intra-tree and inter-tree levels, following Tree-GRPO (Ji et al., 2025). The *Intra-Tree Advantage* $A_{\text{intra}}(v)$ evaluates the relative quality of nodes within the same tree. For a node $v$ in tree $\mathcal{T}^{(i)}$, we standardize $R(v)$

using the mean $\mu_{\text{intra}}^{(i)}$ and standard deviation $\sigma_{\text{intra}}^{(i)}$ derived from that specific tree:

$$
A_{\text{intra}}(v) = \frac{R(v) - \mu_{\text{intra}}^{(i)}}{\sigma_{\text{intra}}^{(i)} + \epsilon}
\tag{11}
$$

Simultaneously, to capture each node's global advantage, we compute the *Inter-Tree Advantage* $A_{\text{inter}}(v)$ against the global mean $\mu_{\text{global}}$ and standard deviation $\sigma_{\text{global}}$ across the entire ensemble $\{\mathcal{T}^{(i)}\}_{i=1}^G$:

$$
A_{\text{inter}}(v) = \frac{R(v) - \mu_{\text{global}}}{\sigma_{\text{global}} + \epsilon}
\tag{12}
$$

The final advantage $A_{\text{total}}(v)$ balances these perspectives:

$$
A_{\text{total}}(v) = A_{\text{intra}}(v) + A_{\text{inter}}(v).
\tag{13}
$$

Through this dual-scale design, the intra-tree advantage supports reliable local comparisons sharing similar contexts and effective credit assignment to identify nodes that critically influence the final outcome. Meanwhile, inter-tree advantages foster competition across different trees, guiding the optimization toward globally high-quality solutions.

**Optimization Objective.** Following the GRPO paradigm, we directly utilize the dual-scale advantage $A_{\text{total}}(v)$ to optimize the retrieval policy $\pi_\theta$ by maximizing:

$$
\mathcal{J}(\theta) = \mathbb{E}_{q \sim \mathcal{D}} \left[ \frac{1}{G} \sum_{i=1}^G \frac{1}{|\mathcal{T}^{(i)}|} \sum_{v \in \mathcal{T}^{(i)}} (\mathcal{L}_{\text{clip}} - \beta \mathbb{D}_{\text{KL}}(\pi_\theta || \pi_{\text{ref}})) \right]
\tag{14}
$$

where $\pi_{\text{ref}}$ constrains the update via the KL penalty coefficient $\beta$. The core term $\mathcal{L}_{\text{clip}}$ applies standard PPO clipping to the probability ratio $\rho_{v,t}(\theta) = \pi_\theta(a_{v,t}|\cdot)/\pi_{\theta_{\text{old}}}(a_{v,t}|\cdot)$:

$$
\mathcal{L}_{\text{clip}} = \min(\rho_{v,t}(\theta) A_{\text{total}}(v), \text{clip}(\rho_{v,t}(\theta), 1 \pm \epsilon) A_{\text{total}}(v))
\tag{15}
$$

### 3.3. MoT-GRPO for Memory Construction

Unlike retrieval, memory construction spans hundreds of steps with rewards delayed until downstream queries, and its quality is irrelevant to most queries, resulting in severe credit assignment ambiguity. Consequently, standard GRPO and Tree-GRPO suffer from excessive online training overhead and severe gradient noise when facing long-horizon dialogues and such sparse dependencies. To address this, we propose **Hindsight Credit Assignment**, which backpropagates advantage signals from downstream retrieval trajectories to upstream construction actions.

**Hindsight Credit Assignment.** Let $a_{\text{mem}}$ be a memory operation processing source turns $\mathcal{X}_{\text{src}}$ to produce a memory entry $m$. For a query $q$ with ground-truth evidence

$\mathcal{X}_{\text{evi}}^q$, we define the hindsight score $S(a_{\text{mem}})$ by aggregating advantages $A_{\text{total}}(v_L)$ from terminal leaf nodes $v_L \in \mathcal{V}_{\text{leaves}}$:

$$S(a_{\text{mem}}) = \frac{1}{|\mathcal{V}_{\text{leaves}}|} \sum_{v_L \in \mathcal{V}_{\text{leaves}}} A_{\text{total}}(v_L) \cdot \varrho(a_{\text{mem}}, v_L) \quad (16)$$

The credit coefficient $\varrho$ integrates two distinct signals:

$$\varrho(a_{\text{mem}}, v_L) = \underbrace{\mathbb{I}(\mathcal{X}_{\text{src}} \cap \mathcal{X}_{\text{evi}}^q \neq \emptyset)}_{\text{Evidence Alignment Gate}} + \lambda \cdot \underbrace{\mathbb{I}(m \in \mathcal{M}_{v_L})}_{\text{Retrieval Trace Gate}}$$
$$(17)$$

The *Evidence Alignment Gate* attributes credit by linking the construction quality of ideal evidence turn $\mathcal{X}_{\text{evi}}^q$ to answer accuracy. It posits that successful reasoning is fundamentally rooted in the effective transformation of ground-truth evidence into memory. Thus, the advantage of a final answer serves as a proxy to evaluate the construction of these pivotal source turns. Conversely, the *Retrieval Trace Gate* (weighted by $\lambda = 0.1$) captures the empirical utility of $m$ retrieved within the actual retrieval tree. It recognizes that any memory entry $m$ involved in the terminal path $\mathcal{M}_{v_L}$ objectively modulates the model's decision-making, rewarding the construction process for its functional contribution to the successful trajectory. Notably, in the absence of ground-truth evidence, the mechanism naturally relies on the *Retrieval Trace Gate*, maintaining robust generalization across diverse datasets.

**Policy Refinement.** To optimize memory construction policies, we employ rank-based sampling to curate a high-quality training dataset $\mathcal{D}_{\text{mem}}^*$. We first discard trajectories with invalid tool invocations. Subsequently, we rank all candidate actions by their hindsight score $S(a_{\text{mem}})$ and retain only the top $50\%$ percentile within each operation category. Finally, treating $\mathcal{D}_{\text{mem}}^*$ as a collection of expert demonstrations, we train the policies $\pi_\theta$ (encompassing $\pi_{\text{form}}$ and $\pi_{\text{evol}}$) to maximize the log-likelihood of these selected actions:

$$\mathcal{L}_{\text{Off}}(\theta) = -\mathbb{E}_{(x,a_{\text{mem}}) \sim \mathcal{D}_{\text{mem}}^*} \left[ \log \pi_\theta(a_{\text{mem}} \mid x, \mathcal{M}_t) \right]. \quad (18)$$

This offline optimization effectively distills the "hindsight wisdom" derived from the downstream MoT-GRPO search trees into the forward-looking memory construction policy.

## 4. Experiments

### 4.1. Experimental Setup

**Evaluation and Benchmarks.** We evaluate the proposed framework across four challenging long-context benchmarks, including LoCoMo (Maharana et al., 2024), Long-MemEval (Wu et al., 2025a), HotpotQA (Yang et al., 2018), and NarrativeQA (Kočiský et al., 2017). LoCoMo and Long-MemEval focus on long-term conversational question answering. Following Memory-R1 (Yan et al., 2025b), we use the same training data configuration by splitting the

LoCoMo dataset into a 1:1:8 train/validation/test split to ensure a fair comparison. The remaining three benchmarks are treated as out-of-domain datasets to evaluate the generalization ability of our method. Specifically, for HotpotQA, following (Yu et al., 2025; Yan et al., 2025a), we construct long-context inputs by concatenating the gold supporting documents with 400 irrelevant Wikipedia documents. More details about the dataset are in Section A.1.

**Baselines.** We compare Mem-T against thirteen baselines, categorized into two groups: **(I) Training-free Methods:** This group includes memory-free approaches, such as vanilla long-LLM and retrieval-augmented generation (RAG) (Lewis et al., 2020), as well as memory-based methods, including MemGPT (Packer et al., 2023), Memory-Bank (Zhong et al., 2024), Mem0 (Chhikara et al., 2025), LightMem (Fang et al., 2025), A-Mem (Xu et al., 2025), and GAM (Yan et al., 2025a). **(II) Training-based Methods:** This group includes MemAgent (Yu et al., 2025) and Mem1 (Zhou et al., 2025), which primarily focus on working memory, and Memory-R1 (Yan et al., 2025b) and Mem-$\alpha$ (Wang et al., 2025), which are designed to mainly enhance factual memory. Based on Mem-T architecture, we compared the training efficacy of GRPO against MoT-GRPO. Furthermore, we discuss Tree-GRPO in Section C.7. For all the baselines, official implementations and released parameters are used when available.

**Implementation Details.** We select LLM backbones of varying sizes, including Qwen3-4B-instruct-2507 and Qwen3-8B (Yang et al., 2025). All methods use BGE-M3 as the embedding model (Chen et al., 2025b). During training with MoT-GRPO, we generate three trees for each query ($G = 3$), with a maximum tree depth of $4$. In each expansion round, we select three nodes ($N_v = 3$) for branch expansion. The training for memory retrieval is conducted for 200 steps. And the training for memory construction is based on a dataset containing $10k$ memory operations. At inference time, Mem-T is allowed up to 6 reasoning steps. All retrieval operations default to returning the top-5 most similar items. More training setup and parameter configurations are listed in Section A.2.

### 4.2. Main Results

**High Performance.** As shown in Table 2 and Table 5, Mem-T achieves substantially better performance on the Lo-CoMo benchmark than both training-free and training-based baselines. When using Qwen3-4B and Qwen3-8B, Mem-T improves F1 by 14.92 (34.13% ↑) and 14.55 (33.08% ↑), respectively. Even without training, the hierarchical and highly agentic memory system of Mem-T achieves superior performance, improving F1 by 5.67 (12.97% ↑) compared to other methods. Moreover, MoT-GRPO further strengthens the LLM's memory management capability compared to

*Table 2.* Performance comparison on the LoCoMo benchmark, with F1 and BLEU-1 as the evaluation metrics. [†]: The GAM paper recommends gpt-4o-mini; we also reproduced it using Qwen3-4B for a fair comparison. [‡]: As Memory-R1 is not open-source, we faithfully report the results provided in their original paper.

| Method | Base LLM | Single-Hop | | Multi-Hop | | Temporal | | Open Domain | | Overall | |
|---|---|---|---|---|---|---|---|---|---|---|---|
| | | F1↑ | B1↑ | F1↑ | B1↑ | F1↑ | B1↑ | F1↑ | B1↑ | F1↑ | B1↑ |
| *Training-free Methods* | | | | | | | | | | | |
| VANILLA | Qwen3-4B | 40.68 | 31.54 | 23.23 | 16.76 | 18.97 | 13.42 | 13.87 | 10.70 | 31.50 | 23.94 |
| RAG | Qwen3-4B | 49.45 | 44.94 | 23.50 | 17.13 | 43.07 | 37.35 | 20.23 | 14.94 | 41.59 | 36.45 |
| MemGPT | Qwen3-4B | 14.00 | 11.77 | 16.68 | 13.99 | 12.56 | 10.94 | 11.61 | 9.16 | 14.05 | 11.84 |
| MemoryBank | Qwen3-4B | 26.65 | 17.72 | 25.52 | 19.44 | 9.15 | 7.44 | 16.42 | 12.39 | 22.34 | 15.66 |
| Mem0 | Qwen3-4B | 47.28 | 40.72 | 35.40 | 27.36 | 46.84 | 39.48 | 26.64 | 21.04 | 43.71 | 36.78 |
| MemoryOS | Qwen3-4B | 48.35 | 42.57 | 35.24 | 27.30 | 40.98 | 32.68 | 22.08 | 17.93 | 42.83 | 36.26 |
| LightMem | Qwen3-4B | 43.78 | 38.84 | 30.78 | 25.80 | 44.71 | 40.72 | 18.93 | 14.42 | 40.01 | 35.27 |
| A-Mem | Qwen3-4B | 44.62 | 38.26 | 27.24 | 21.07 | 43.85 | 35.97 | 15.40 | 12.71 | 39.43 | 33.04 |
| GAM | Qwen3-4B | 32.23 | 25.54 | 32.23 | 28.66 | 26.26 | 22.52 | 18.45 | 14.47 | 30.17 | 24.81 |
| GAM[†] | gpt-4o-mini[†] | 57.75 | 52.10 | 42.29 | 34.44 | 59.45 | 53.11 | 29.73 | 24.74 | 53.48 | 47.33 |
| *Trained Methods* | | | | | | | | | | | |
| MEM1 | MEM1-7B | 27.48 | 22.10 | 18.98 | 15.56 | 30.52 | 23.48 | 14.21 | 11.43 | 25.68 | 20.50 |
| MemAgent | MemAgent-14B | 35.86 | 29.64 | 27.86 | 22.72 | 37.93 | 31.85 | 20.31 | 16.47 | 33.82 | 27.97 |
| Memory-R1-PPO[‡] | Mem-R1-8B[‡] | 32.52 | 24.47 | 26.86 | 23.47 | 41.57 | 26.11 | 45.30 | 39.18 | 34.08 | 25.54 |
| Memory-R1-GRPO[‡] | Mem-R1-8B[‡] | 35.73 | 27.70 | 35.65 | 30.77 | 49.86 | 38.27 | **47.42** | **41.24** | 39.25 | 31.21 |
| *Our Method* `Mem-T` | | | | | | | | | | | |
| *w/o* training | Qwen3-4B | 53.97 | 49.15 | 38.44 | 31.70 | 53.99 | 48.08 | 26.44 | 23.37 | 49.38 | 44.11 |
| with GRPO | Qwen3-4B | 59.43 | 54.65 | 38.40 | 30.51 | 60.78 | 56.10 | 23.46 | 20.16 | 53.56 | 48.33 |
| with MoT-GRPO | Qwen3-4B | **63.75** | **57.95** | **45.09** | **36.58** | **65.13** | **60.12** | 32.97 | 28.94 | **58.65** | **52.63** |

*Table 3.* Evaluation results on OOD benchmarks (HotpotQA, Long-MemEval, NarrativeQA). All methods, except MEM1, which uses the 7B model trained in the original paper, are implemented with models based on Qwen3-4B.

| Method | HotpotQA F1↑ | LongMemEval Acc↑ | NarrativeQA F1↑ | Avg. |
|---|---|---|---|---|
| *Training-free Methods* | | | | |
| VANILLA | 21.89 | 38.80 | 18.09 | 26.26 |
| RAG | 50.13 | 56.60 | 21.17 | 42.63 |
| MemGPT | 18.24 | 23.00 | 8.39 | 16.54 |
| MemoryBank | 16.90 | 26.20 | 9.65 | 17.58 |
| A-Mem | 30.46 | 61.30 | 25.18 | 38.98 |
| Mem0 | 31.96 | 53.60 | 27.63 | 37.73 |
| MemoryOS | 26.86 | 46.80 | 23.45 | 32.37 |
| LightMem | 38.62 | 63.10 | 16.78 | 39.50 |
| GAM | 52.98 | 61.80 | 28.32 | 47.70 |
| *Trained Methods* | | | | |
| MEM1 | 55.36 | 19.00 | 13.49 | 29.28 |
| Mem-$\alpha$ | 58.80 | 52.00 | 28.56 | 46.45 |
| `Mem-T` | **66.35** | **65.80** | **30.29** | **54.15** |

the training-free and the GRPO baseline, yielding additional F1 gains of 9.27 (18.77% ↑) and 5.09 (9.50% ↑). These results demonstrate that the joint retrieval and construction training with dense rewards in MoT-GRPO is better suited for long-horizon memory agents. Notably, GAM, the SOTA memory system, exhibits an F1 gap of 23.31 when switching its backbone from gpt-4o-mini to Qwen3-4B, highlighting the importance of systematically improving model-level memory management capabilities.

**Cross-domain generalization.** To evaluate whether the memory management capabilities learned by MoT-GRPO can transfer across tasks, we assess the performance of `Mem-T` on three out-of-domain tasks. As shown in Table 3, baselines such as LightMem achieve suboptimal performance on LongMemEval but fail to generalize to other benchmarks, trailing `Mem-T` by 27.73 and 13.51 on HotpotQA and NarrativeQA, respectively. Training-based MEM-1 performs well on the in-domain QA benchmark HotpotQA, outperforming training-free methods by 2.38, but suffers substantial degradation on benchmarks that emphasize long-horizon dialogue understanding, underperforming `Mem-T` by 46.8 and 16.8. In contrast, `Mem-T` learns effective memory management strategies through training on LoCoMo and achieves SOTA performance across all three out-of-domain benchmarks, with an average improvement of 6.45(13.52% ↑) over other methods. Notably, `Mem-T` generalizes well from long-horizon dialogue to the QA setting of HotpotQA, outperforming other approaches by 7.55.

**Token-economical.** As illustrated in Figure 3 and Figure 6, `Mem-T` demonstrates superior cost-effectiveness, lying on the Pareto front for both the LoCoMo and HotpotQA datasets. Compared to GAM, `Mem-T` not only achieves a

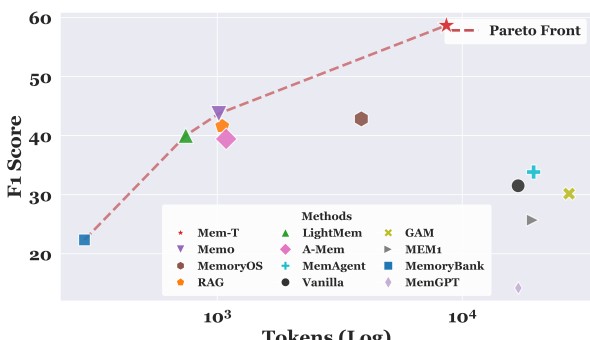

Figure 3. The comparison of the performance and inference cost on the LoCoMo dataset. Different shapes of the scatter points represent various types of baselines.

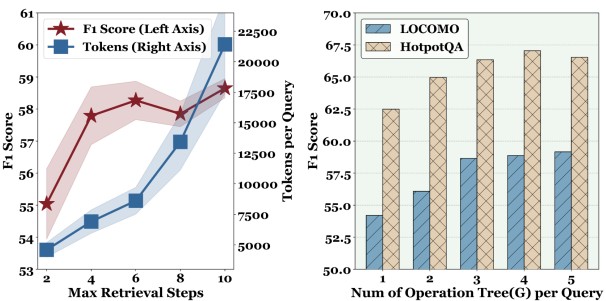

Figure 4. (**Left**) Parameter sensitivity analysis on the max inference retrieval steps on the LoCoMo; (**Right**) Parameter sensitivity analysis on the number of operation trees per query($G$) when training with MoT-GRPO on the LoCoMo and HotpotQA dataset.

Table 4. Ablation study on the LoCoMo dataset. The evaluation metric is set as F1 for all entries.

| Method | Single | Multi | Temporal | Open | Overall |
|---|---|---|---|---|---|
| Vanilla `Mem-T` | 63.75 | 45.09 | 65.13 | 32.97 | 58.65 |
| *Ablation of Memory Modules* | | | | | |
| w/o $\mathcal{M}_{\text{work}}$ | 63.24 | 43.42 | 63.38 | 30.90 | 57.59 |
| w/o $\mathcal{M}_{\text{fact}}$ | 60.80 | 40.10 | 64.23 | 22.39 | 55.25 |
| w/o $\mathcal{M}_{\text{exp}}$ | 61.94 | 43.96 | 62.64 | 27.42 | 56.60 |
| w/o $\mathcal{M}_{\text{raw}}$ | 62.19 | 42.84 | 62.41 | 29.38 | 56.61 |
| *Ablation of MoT-GRPO* | | | | | |
| w/o Retr. Opt. | 57.91 | 43.85 | 56.69 | 30.73 | 53.37 |
| w/o Cons. Opt. | 61.41 | 41.15 | 61.17 | 25.24 | 55.36 |
| w/o $A_{\text{intra}}$ | 62.08 | 43.08 | 63.52 | 31.53 | 56.95 |
| w/o $A_{\text{inter}}$ | 58.33 | 43.52 | 59.58 | 30.59 | 54.09 |

$5.17 \sim 28.48$ improvement in F1 Score but also reduces the inference overhead by $19.94\% \sim 24.45\%$ per query. A detailed analysis of the training overhead of MoT-GRPO is provided in Section B.1.

### 4.3. Framework Analysis

**Ablation Study** We conduct an ablation study on the hierarchical memory architecture and the MoT-GRPO training paradigm, with results presented in Table 4 and Section C.3: **(1) w/o Memory Modules**, which individually removes the working ($\mathcal{M}_{\text{work}}$), factual ($\mathcal{M}_{\text{fact}}$), experiential ($\mathcal{M}_{\text{exp}}$), and raw ($\mathcal{M}_{\text{raw}}$) memory stores. On LoCoMo, which emphasizes information extraction in long-horizon dialogues, factual memory proves to be the most critical component, leading to a substantial performance decline of $3.40$. **(2) w/o Optimization Strategies**, where we replace the MoT-GRPO-optimized policies with the base model during the memory retrieval (*w/o* Retr. Opt.) and construction (*w/o* Cons. Opt.) phases. Eliminating the retrieval optimization leads to the most significant performance decline of $5.28$, while removing the construction optimization causes a $3.29$ drop. These marked degradations verify that both stages of

MoT-GRPO are crucial. **(3) w/o Advantage Terms**, which ablates the intra-tree ($A_{\text{intra}}$) or inter-tree ($A_{\text{inter}}$) advantage. Removing $A_{\text{inter}}$ causes a larger performance drop ($4.56 \downarrow$) than removing $A_{\text{intra}}$ ($1.70 \downarrow$), indicating that cross-tree advantage estimation is critical for stable RL training, while combining both signals yields the best performance.

**Sensitivity Analysis** We analyze the sensitivity of `Mem-T` to three core parameters. The results are presented in Figure 4 and Figure 7. **For the maximum retrieval steps**, we observe a substantial performance improvement as the steps increase from 2 to 6, where the F1 score increases from $53.45 \rightarrow 58.65$. However, further extending the steps from 6 to 10 yields only marginal gains ($< 0.5\%$) while linearly inflating the token consumption per query from $\sim 9k$ to $\sim 21k$. **For the number of operation trees** $G$, increasing $G$ from 1 to 3 yields substantial gains, boosting the F1 score on LoCoMo from $54.20$ to $58.65$ and on HotpotQA from $62.49$ to $66.54$. However, further increasing $G$ to 5 results in diminishing returns, offering a marginal average improvement of only $0.35$ while disproportionately inflating the computational cost by approximately $67\%$. Thus, we set the maximum retrieval steps to 6 and $G = 3$ to balance efficiency and overhead. More analysis is in Section C.4.

### 4.4. Case Study

We present a case study comparing the memory processing trajectories of `Mem-T` against the Qwen3-4B baseline in Figure 5 to demonstrate the enhanced capabilities acquired through our training paradigm.

As illustrated, the baseline exhibits severe limitations across the entire memory lifecycle. In the **formation** phase, it lacks an accurate information extraction capability, failing to resolve relative timestamps (e.g., "yesterday") into specific dates. During **evolution**, it fails to distinguish between `Update` and `Add` operations, erroneously overwriting existing entity records with unrelated new memory. Finally, its **retrieval** mechanism is limited to ambiguous raw queries,

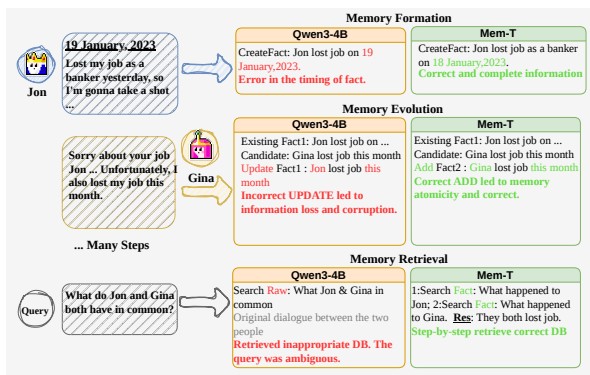

*Figure 5.* Case Study comparing `Mem-T` against baseline.

lacking the logical depth to handle multi-step reasoning.

In contrast, `Mem-T` demonstrates superior capabilities in three aspects:❶**Accurate Information Extraction:** It accurately processes raw information (e.g., converting "yesterday" to a correct specific date), ensuring initial memory entries are temporally grounded and factually complete; ❷ **Rational Memory Evolution:** It exhibits a deep understanding of the usage criteria for memory evolution tools. By explicitly distinguishing between state updates and new knowledge acquisition, it preserves memory atomicity and prevents key information forgetting. ❸ **Multi-step Retrieval:** Instead of vague searches, it autonomously decomposes complex queries into sub-questions and retrieves from a suitable store. This step-by-step memory lookups synthesize the answer from distinct memory entries.

## 5. Conclusion

In this paper, we introduce `Mem-T`, a comprehensive hierarchical memory framework, and MOT-GRPO, a novel RL paradigm that pioneers the application of tree-based RL to memory retrieval training, while introducing a memory-pivoted credit assignment for memory construction training. By propagating sparse terminal rewards into dense, step-wise supervision via memory operation trees, MOT-GRPO enables the joint optimization of memory construction and retrieval policies. The extensive experiments demonstrate that `Mem-T` not only achieves state-of-the-art performance across in-domain and out-of-domain benchmarks but also realizes a superior Pareto efficiency between task accuracy and inference overhead. We believe `Mem-T` represents a shift from heuristic-based storage to fully learnable, attribution-centric memory systems, paving the way for the development of self-evolving agents capable of lifelong learning.

## Acknowledgments

This research is supported in part by Ucap Cloud.

This research is part of the IN-CYPHER programme and is supported by the National Research Foundation, Prime Minister's Office, Singapore under its Campus for Research Excellence and Technological Enterprise (CREATE) programme.

## Impact Statement

This paper presents work whose goal is to advance the field of Machine Learning. All experiments are conducted on publicly available benchmarks, which do not contain sensitive personal information. We do not intentionally collect, infer, or generate content that identifies specific individuals. We feel that no other potential societal consequences of our work must be specifically highlighted here.

While our findings demonstrate the efficacy of `Mem-T`, several avenues remain open for future investigation. Currently, our evaluations are focused on static long-context dialogue benchmarks utilizing relatively compact base models (e.g., 4B and 8B parameters). To unlock the full potential of this framework, subsequent efforts will explore scaling up the underlying foundation models and deploying them in more complex, long-horizon agent environments. In terms of efficiency, although `Mem-T` incurs relatively low overhead during inference and training for specific questions, the overhead for memory formation and update operations on long contexts is comparatively high. Furthermore, while `Mem-T` achieves its optimal performance when assisted by gold evidence labels, this dependency highlights a compelling opportunity to develop automated, self-supervised annotation algorithms.

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

# A. Supplementary Experimental Setup

## A.1. Dataset Description

**LoCoMo((Maharana et al., 2024))** is a benchmark of very long-term conversational dialogues designed to evaluate long-range memory and reasoning capabilities in agent systems. The dataset consists of 10 extended conversations, each spanning dozens of sessions and hundreds of dialogue turns, with an average of around 600 turns and roughly 16K tokens per conversation. Questions in the LoCoMo QA evaluation are annotated with answer locations and categorized into types such as single-hop, multi-hop, open-domain, temporal reasoning, and adversarial, targeting different memory and inference challenges. In our experiments on LoCoMo QA, we follow standard practice in related work and do not use adversarial question data, which aligns with previous evaluations (Chhikara et al., 2025; Xu et al., 2025).

**Hotpotqa((Yang et al., 2018))** is a widely-used multi-hop reasoning benchmark that requires models to aggregate information across multiple supporting documents to reach an answer. To evaluate performance in long-context scenarios, we follow the synthesis methodology proposed in recent work (Yu et al., 2025; Yan et al., 2025a), where the golden paragraphs containing the necessary evidence are embedded within a haystack of distractor content. In our experiments, we specifically utilize the 56K-token(eval_400) version of this synthetic HotpotQA dataset. This setup effectively transforms the reasoning task into a long-range retrieval and inference challenge, testing the agent's ability to filter out extensive irrelevant information while maintaining the precision required for multi-step logical reasoning.

**LoneMemEval((Wu et al., 2025a))** is specifically designed to evaluate the long-term interactive memory capabilities of LLM-driven chat assistants, addressing the underexplored challenge of sustained memory performance in prolonged user-AI interactions. It comprehensively assesses five core memory abilities, information extraction, multi-session reasoning, temporal reasoning, knowledge updates, and abstention, through 500 manually curated questions embedded in freely scalable user-assistant chat histories, with two standard configurations: LONGMEMEVALS ($115k$ tokens per question) and LONGMEMEVALM ($\sim 1.5$ million tokens across 500 sessions). Following previous works (Wang et al., 2025; Rasmussen et al., 2025; Fang et al., 2025), we use the LONGMEMEVALS dataset.

**NarrativeQA((Kočiský et al., 2017))** is a large-scale reading comprehension benchmark that assesses models' ability to understand and reason over long narrative text, such as books and movie scripts. The full NarrativeQA dataset contains on the order of tens of thousands of human-written question–answer pairs associated with over a thousand story documents, where questions require synthesis across global document structure rather than shallow pattern matching. Questions are constructed based on human-generated abstractive summaries, encouraging deep narrative understanding and integrative reasoning beyond local context overlaps. Following (Hu et al., 2026a), we randomly sampled 10 long documents from the NarrativeQA corpus and used their associated 298 QA pairs to measure performance on long-range narrative question answering.

## A.2. Implementation Details

**MoT-GRPO for Memory Retrieval** *Training Implementation Details.* We utilize the Ray distributed framework combined with vLLM as the inference backend, employing XFormers to optimize attention mechanisms. The model is trained with a global batch size of 32. We adopt a peak learning rate of $5 \times 10^{-6}$ with a warmup ratio of $0.285$. To ensure training stability and prevent reward hacking, we set the KL divergence coefficient to $0.001$.

*Context and Efficiency.* To support extensive memory retrieval operations, we configure the system with an extended context window, allowing for a maximum prompt length of 40,960 tokens and a maximum observation history of 20,480 tokens. For computational efficiency, we employ Fully Sharded Data Parallel (FSDP) with parameter, gradient, and optimizer offloading, performing all computations in `bfloat16` precision.

**MoT-GRPO for Memory Construction** *Training Configuration.* The training is conducted on a single node equipped with 8 GPUs, utilizing the LLaMA-Factory framework. To maximize computational efficiency and handle the memory footprint of full-parameter updates, we employ DeepSpeed ZeRO-3 combined with Flash Attention 2. The maximum sequence length is truncated to $6,144$ tokens.

*Hyperparameters.* The global batch size is set to 32 (calculated with a per-device batch size of 2 and 2 gradient accumulation steps). We optimize the model for 200 steps using a cosine learning rate scheduler, with a peak learning rate of $5 \times 10^{-6}$

and a warmup ratio of 0.1. The training uses `bfloat16` precision, and $10\%$ of the dataset is reserved for validation to monitor convergence.

## B. Theoretical Analysis

### B.1. Efficiency Analysis of Training Overhead

We provide an analysis comparing the efficiency of MoT-GRPO to standard GRPO in terms of token generation and memory consumption during memory retrieval training and memory construction training.

**1. Retrieval Training Token Efficiency**    Let $G$ denote the number of initial trees, $K$ be the number of branches, and $L$ represent the average trajectory length. The total number of trajectories generated is $G(1 + K)$.

- **GRPO Cost:** Generating all trajectories independently incurs a token cost of:

$$\text{Cost}_{\text{GRPO}} = G(1 + K) \cdot L$$

- **MoT-GRPO Cost:** Generating the $G$ initial full trajectories costs $G \cdot L$. For the new branches, the prefix sequence is reused. Assuming the branching occurs halfway through, the average length of the generated branch is $L/2$, resulting in a branching cost of $G \cdot K \cdot L/2$. The total cost is therefore:

$$\text{Cost}_{\text{MoT-GRPO}} = G \cdot L + G \cdot K \cdot \frac{L}{2}$$

Comparing the two costs, we obtain the efficiency ratio:

$$\text{Ratio} = \frac{G \cdot L + G \cdot K \cdot L/2}{G(1 + K) \cdot L} = \frac{1 + K/2}{1 + K} \approx 0.5$$

Consequently, `Mem-T` **generates the same number of rollouts as GRPO at approximately half the token cost.**

**2. Construction Training Memory Efficiency**    We analyze the memory requirements using Qwen3-4B (FP16, hidden dimension $= 2560$, 36 layers) processing a 40k-token dialogue (resulting in a $\sim$60k construction context). By employing FlashAttention-2 and full gradient checkpointing, the activation cost per rollout is calculated as:

$$2 \times 60\text{k} \times 2560 \times 36 \approx 10.30 \text{ GB}$$

With a standard group size of 16, GRPO incurs a massive activation overhead of approximately 164.79 GB per query.

In contrast, by shifting to offline memory construction training, MoT-GRPO reduces the number of concurrent trajectories from 16 to 1. This strategic shift **cuts the activation memory requirement from $\sim$165 GB down to $\sim$10 GB**. Furthermore, it eliminates the need to load a reference model into memory, effectively reducing the overall computational cost to the level of Supervised Fine-Tuning.

### B.2. Proving MoT-GRPO's Lower Gradient Variance for Memory Retrieval Training

Consider an intermediate node $v_t = \langle a_t, h_t \rangle$ with the score function defined as $s_t = \nabla_\theta \log \pi_\theta(a_t \mid h_t)$. The per-step policy gradient estimators for standard GRPO and MoT-GRPO are given by:

$$g_t^{\text{GRPO}} = s_t \cdot \hat{A}_t^{\text{GRPO}}, \quad g_t^{\text{MoT}} = s_t \cdot \hat{A}_t^{\text{MoT}} \tag{19}$$

Assume each branch is independently sampled given the history $h_t$, and the terminal rewards $R(\tau)$ are conditionally independent and identically distributed (i.i.d.) with a variance of $\text{Var}[R(\tau) \mid h_t, a_t] = \sigma^2$.

For GRPO, the conditional variance is straightforward:

$$\text{Var}[R_t^{\text{GRPO}} \mid h_t, a_t] = \sigma^2 \tag{20}$$

For MoT-GRPO, we expand the node reward as $R(v_t) = \alpha \cdot \text{Evid}(v_t) + \text{Perform}(v_t)$. The variance of this node reward is:

$$\text{Var}[R(v_t) \mid h_t, a_t] = \alpha^2 \cdot \underbrace{\text{Var}[\text{Evid}(v_t) \mid h_t, a_t]}_{=0 \text{ (deterministic given } a_t)} + \text{Var}[\text{Perform}(v_t) \mid h_t, a_t]$$
$$= \text{Var}[\text{Perform}(v_t) \mid h_t, a_t] \tag{21}$$

For the performance term, applying the variance formula for weighted sums of independent leaf rewards yields:

$$\text{Var}[\text{Perform}(v_t) \mid h_t, a_t] = \sigma^2 \cdot \sum_{v_L \in \mathcal{L}(v_t)} w_{v_L}^2 \tag{22}$$

By Jensen's inequality (or more simply, the properties of convex combinations), when the number of leaves $|\mathcal{L}(v_t)| \geq 2$ and all weights satisfy $w_{v_L} > 0$:

$$\sum_{v_L} w_{v_L}^2 < \left( \sum_{v_L} w_{v_L} \right)^2 = 1 \tag{23}$$

Therefore, it follows that:

$$\text{Var}[R(v_t) \mid h_t, a_t] < \sigma^2 = \text{Var}[R_t^{\text{GRPO}} \mid h_t, a_t] \tag{24}$$

Propagating this to the gradient variance (since $s_t$ is deterministic given the state-action pair $(h_t, a_t)$), we obtain:

$$\text{Var}[g_t^{\text{MoT}}] < \text{Var}[g_t^{\text{GRPO}}] \tag{25}$$

In conclusion, the evidence-hit term acts as a zero-variance anchor, and the weighted averaging of child performance further reduces variance relative to any single leaf outcome. Together, these two mechanisms equip MoT-GRPO with a provably **lower-variance gradient estimator for retrieval training**.

## B.3. How Densified Rewards Improve Signal-to-Noise Ratio (SNR) for Memory Construction Training

We formalize the benefit of hindsight credit assignment by analyzing the gradient Signal-to-Noise Ratio (SNR), defined as:

$$\text{SNR}(\hat{g}) = \frac{\|\mathbb{E}[\hat{g}]\|_2}{\sqrt{\text{Tr}(\text{Var}(\hat{g}))}} \tag{26}$$

Consider a long context containing $N$ construction operations, where only a causal subset of size $k \ll N$ is relevant to any given downstream answer. Under a standard sparse terminal reward framework that broadcasts uniformly to all $N$ operations, the SNR is:

$$\text{SNR}_{\text{sparse}} = \frac{\mathcal{O}(k)}{\sqrt{\mathcal{O}(N \cdot \text{Var}(R_{\text{term}}))}} = \mathcal{O}\left( \frac{k}{\sqrt{N}} \right) \xrightarrow{N \to \infty} 0 \tag{27}$$

As the context length $N$ grows, the gradient degrades into pure noise.

Conversely, our Hindsight Credit Assignment (HCA) mechanism densifies rewards via the Retrieval Trace Gate. This gate applies an indicator mask $\mathbb{I}(m \in \mathcal{M}_{v_L})$. Consequently, for the $N - k$ irrelevant operations, the hindsight score is exactly zero ($S(a_i) = 0$). The gradient formulation is thereby restricted to only the $k$ relevant actions within subset $\mathcal{K}$:

$$\hat{g}_{\text{dense}} = \frac{1}{k} \sum_{i \in \mathcal{K}} \nabla_\phi \log \pi_\phi(a_i) S(a_i) \tag{28}$$

Because the noise variance is now strictly bounded by $k$ rather than $N$, the resulting SNR becomes:

$$\text{SNR}_{\text{dense}} = \frac{\mathcal{O}(k)}{\sqrt{\mathcal{O}(k \cdot \text{Var}(S))}} = \mathcal{O}(\sqrt{k}) \tag{29}$$

Crucially, this is $\mathcal{O}(1)$ with respect to the total sequence length $N$. **HCA truncates the covariance trace from $\mathcal{O}(N)$ down to $\mathcal{O}(k)$**, effectively decoupling the SNR from the overall horizon length and guaranteeing robust credit assignment regardless of the context length.

# C. Supplementary Experiment

## C.1. Generalization Experiments Across Other LLMs

*Table 5.* Performance comparison on the LoCoMo benchmark, with F1 and BLEU-1 as the evaluation metrics. [‡]: As Memory-R1 is not open-source, we faithfully report the results provided in their original paper.

| Method | LLM | Single-Hop | | Multi-Hop | | Temporal | | Open Domain | | Overall | |
|---|---|---|---|---|---|---|---|---|---|---|---|
| | | F1↑ | B1↑ | F1↑ | B1↑ | F1↑ | B1↑ | F1↑ | B1↑ | F1↑ | B1↑ |
| *Training-free Methods* | | | | | | | | | | | |
| VANILLA | Qwen3-8B | 38.55 | 33.91 | 30.66 | 21.94 | 25.14 | 20.73 | 17.80 | 14.04 | 33.14 | 27.86 |
| RAG | Qwen3-8B | 49.62 | 43.98 | 23.64 | 17.82 | 37.93 | 33.80 | 21.39 | 16.33 | 40.77 | 35.43 |
| MemGPT | Qwen3-8B | 16.23 | 13.08 | 18.13 | 13.72 | 15.87 | 11.39 | 14.18 | 10.66 | 16.38 | 12.71 |
| MemoryBank | Qwen3-8B | 26.50 | 19.48 | 26.52 | 18.93 | 15.49 | 11.36 | 15.92 | 12.09 | 23.66 | 17.31 |
| Mem0 | Qwen3-8B | 45.92 | 39.93 | 27.80 | 19.97 | 43.64 | 33.82 | 18.37 | 13.84 | 40.41 | 33.42 |
| MemoryOS | Qwen3-8B | 48.77 | 43.47 | 29.19 | 24.87 | 42.98 | 35.27 | 18.50 | 15.09 | 42.12 | 36.65 |
| LightMem | Qwen3-8B | 49.89 | 44.48 | 33.98 | 27.60 | 44.53 | 39.65 | 19.37 | 14.05 | 43.98 | 38.51 |
| A-Mem | Qwen3-8B | 47.75 | 41.36 | 32.35 | 24.82 | 36.80 | 30.71 | 18.62 | 14.98 | 40.92 | 34.56 |
| GAM | Qwen3-8B | 46.62 | 40.15 | 32.18 | 24.96 | 46.42 | 39.71 | 13.56 | 10.32 | 41.84 | 35.39 |
| *Trained Methods* | | | | | | | | | | | |
| MEM1 | MEM1-7B | 27.48 | 22.10 | 18.98 | 15.56 | 30.52 | 23.48 | 14.21 | 11.43 | 25.68 | 20.50 |
| MemAgent | MemAgent-14B | 35.86 | 29.64 | 27.86 | 22.72 | 37.93 | 31.85 | 20.31 | 16.47 | 33.82 | 27.97 |
| Memory-R1-PPO[‡] | Mem-R1-8B | 32.52 | 24.47 | 26.86 | 23.47 | 41.57 | 26.11 | 45.30 | 39.18 | 34.08 | 25.54 |
| Memory-R1-GRPO[‡] | Mem-R1-8B | 35.73 | 27.70 | 35.65 | 30.77 | 49.86 | 38.27 | **47.42** | **41.24** | 39.25 | 31.21 |
| *Our Method* | | | | | | | | | | | |
| w/o training | Qwen3-8B | 55.89 | 51.14 | 38.13 | 30.33 | 53.30 | 47.02 | 23.55 | 20.18 | 50.08 | 44.55 |
| with MoT-GRPO | Qwen3-8B | **63.65** | **57.97** | **42.38** | **34.72** | **66.85** | **62.29** | 34.33 | 31.47 | **58.53** | **52.89** |

Table 5 demonstrates the generalization capabilities of `Mem-T` when applied to the Qwen3-8B model. The results indicate that our approach significantly outperforms all existing baselines across most metrics in the LoCoMo benchmark. Notably, even our training-free variant achieves an Overall F1 score of 50.08, surpassing previously established trained models such as Memory-R1 and MemAgent. When combined with our specific training, the performance further improves to 58.53 F1, particularly excelling in Single-Hop and Temporal reasoning tasks, thereby confirming the robust transferability and effectiveness of our framework across different LLM backbones.

## C.2. Pareto Front Analysis

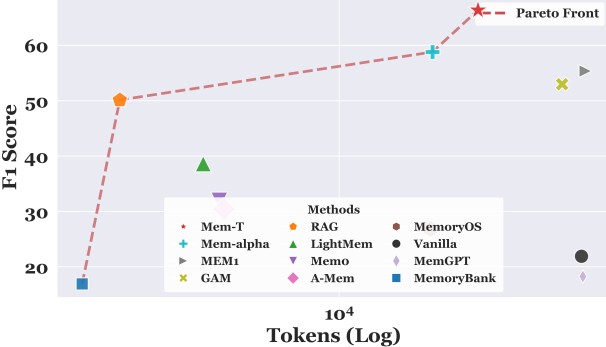

*Figure 6.* The comparison of the performance and inference cost on the HotpotQA dataset. Different shapes of the scatter points represent various types of baselines.

As illustrated in Figure 6, `Mem-T` demonstrates superior cost-effectiveness, lying on the Pareto front for HotpotQA datasets.

## C.3. Ablation Study

To dissect the empirical contributions of individual components within `Mem-T`, we evaluate the core algorithmic pillars: the tree search mechanism and the evidence alignment gate derived from gold evidence signals. As illustrated in Table 6, removing the tree search mechanism (*w/o tree search*) leads to a severe performance degradation, with F1 and B1 scores dropping by $2.78\%$ and $4.05\%$, respectively. This underscores the necessity of structural exploration over heuristic-based retrieval. Furthermore, excluding the evidence-driven advantage signal (*w/o Evidence*) yields a notable performance drop, validating its efficacy in mitigating sparse reward issues.

*Table 6.* Ablation results on the LoCoMo dataset. *Mem-T* denotes our full framework.

| Method | F1 | B1 |
|---|---|---|
| **Mem-T (Full)** | **58.65** | **52.63** |
| *w/o* tree search | 55.87 | 48.58 |
| *w/o* Evidence | 56.32 | 50.74 |

Although omitting the evidence signals leads to a performance degradation, `Mem-T` still consistently outperforms the strongest baseline, thereby demonstrating the robust overall efficacy of our proposed framework.

## C.4. Sensitivity Analysis

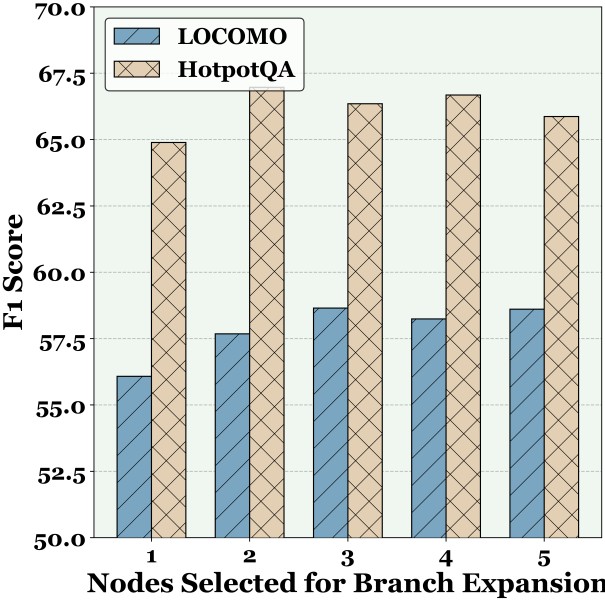

*Figure 7.* Parameter sensitivity analysis on the number of nodes selected for branch expansion when training with MOT-GRPO on the LoCoMo and HotpotQA dataset.

**Regarding the number of nodes selected for branch expansion**, as shown in Figure 7, we observe that increasing the number of nodes selected for branch expansion from 1 to 3 leads to significant performance improvements, with the F1 score rising from 56.08 to 58.65 on LoCoMo and from 64.89 to 66.35 on HotpotQA. However, further increasing the expansion breadth beyond 3 nodes yields diminishing returns; for instance, at a node count of 5, the F1 scores for both datasets plateau or even slightly decrease. Given that a larger number of expansion nodes significantly increases the search space and computational latency, we select 3 as the optimal number of nodes for branch expansion to achieve the best trade-off between reasoning accuracy and inference efficiency.

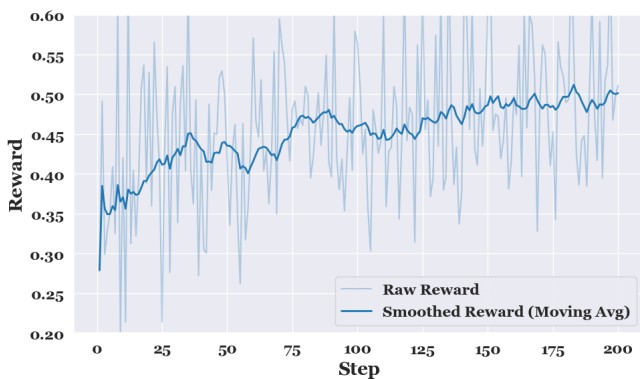

*Figure 8.* Reward curves of memory retrieval training under MOT-GRPO.

## C.5. Training Curves

**Regarding the memory retrieval training stage**, Figure 8 illustrates the evolution of rewards under the MOT-GRPO framework. The smoothed reward curve exhibits a consistent upward trend, climbing from an initial value of approximately $0.30$ to over $0.50$ by the 200th step. Although the raw rewards show significant variance, typical of reinforcement learning in complex reasoning tasks, the steady improvement in the moving average confirms that the agent effectively learns to optimize its retrieval strategies to maximize task-specific gains.

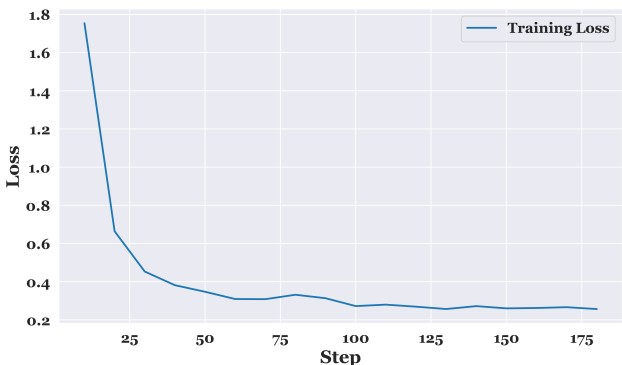

*Figure 9.* Loss curves of memory construction training under MOT-GRPO.

**For the memory construction training stage**, Figure 9 presents the training loss over 180 steps. The curve shows a sharp initial descent, with the loss dropping rapidly from $1.8$ to below $0.4$ within the first 50 steps, indicating efficient convergence. In the subsequent phase, the loss stabilizes and fluctuates marginally around $0.25$, suggesting that the model has successfully captured the underlying patterns for memory synthesis and state updates. The overall stability of the loss curve demonstrates the robustness of the memory construction process under our policy optimization framework.

## C.6. Quantitative Node Importance Analysis

To validate the efficacy of our memory operation scoring, we conducted a quantitative analysis on 5,646 memory construction nodes. Specifically, we examined the correlation between the assigned hindsight advantages of these nodes and the mean F1 scores of their corresponding queries. The results, illustrated in Figure 10, demonstrate extremely strong statistical significance, yielding a Pearson correlation coefficient of $r = 0.867$ and a Spearman rank correlation coefficient of $\rho = 0.926$ ($p < 0.0001$). This near-perfect rank correlation quantitatively confirms that our framework accurately assigns the highest importance scores to the memory operations that are most critical in driving successful final answers.

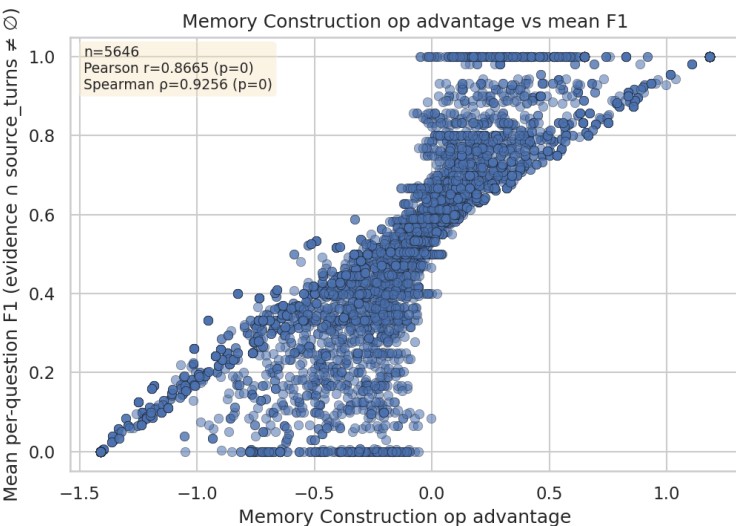

*Figure 10.* Scatter plot illustrating the correlation between hindsight advantages and mean F1 for memory construction nodes.

## C.7. Comparing MOT-GRPO with Tree-GRPO

While both our framework and Tree-GRPO leverage tree-structured search and dual-scale advantage estimation to guide reinforcement learning, our approach fundamentally differs in architectural design, optimization efficiency, and credit assignment granularity.

**Methodological Differences** We outline two core distinctions below:

- **Tree Construction Strategy:** Tree-GRPO constructs search trees online across all operations. In contrast, to prevent combinatorial explosion, our framework decouples the temporal horizons: we employ active tree search exclusively for short-horizon retrieval operations, while long-horizon memory construction is optimized via offline hindsight credit assignment.

- **Advantage Estimation Granularity:** Tree-GRPO computes advantages at the global trajectory level and subsequently redistributes them to intermediate nodes. Our framework computes advantages directly at the node level by combining format correctness, evidence hit rate, and a gated credit assignment mechanism.

**Mitigating Scalability and Sparsity Bottlenecks** The monolithic online tree training of Tree-GRPO encounters severe bottlenecks when applied to long-term memory tasks:

- **Computational Scalability:** Online tree search for long sequences ($> 100$ construction operations) is computationally prohibitive due to exponential branching. For instance, scaling to a 60k context window with a batch size of 16 using a Qwen3-4B backbone incurs approximately $165\,\mathrm{GB}$ of activation memory. Our decoupled two-stage training bypasses this entirely.

- **Sparse Relevance Gradient Noise:** In memory-augmented generation, downstream rewards typically depend on a small fraction ($\approx 5\%$) of relevant memory entries. Tree-GRPO's layer-wise advantage backpropagation distributes non-zero credits across all nodes, injecting massive gradient noise into the remaining $95\%$ irrelevant constructions. Our node-level attribution strictly isolates credit to textually and logically relevant nodes.

Consequently, rather than a trivial extension of Tree-GRPO, our framework introduces a generalized memory architecture that encompasses diverse memory types and operations. Specifically, it employs a Memory Operation Tree for online training of memory retrieval, while leveraging hindsight credit assignment for the offline training of memory construction.

**Empirical Evaluation**    To ensure a fair empirical comparison under identical compute constraints, we evaluated a retrieval-tuned variant of Tree-GRPO on the LoCoMo benchmark, restricting its optimization exclusively to the retrieval phase. As demonstrated in Table 7, our retrieval-tuned baseline outperforms Tree-GRPO, while our full framework (*Ours (Full)*) yields substantial gains across all metrics.

*Table 7.* Performance comparison against Tree-GRPO on the LoCoMo dataset. All retrieval-tuning variants are evaluated under constrained training costs.

| Method | F1 | B1 |
|---|---|---|
| Tree-GRPO (Retrieval-tuning) | 54.20 | 47.92 |
| Ours (Retrieval-tuning) | 55.36 | 49.51 |
| **Ours (Full)** | **58.65** | **52.63** |

### C.8. Fair comparison using the identical base model.

A common confounding factor in evaluating memory-augmented frameworks is the variance in the underlying Large Language Model (LLM) capacity, which can inadvertently obscure the true algorithmic contributions. To eliminate this backbone bias and isolate the empirical gains of our architectural design, we establish an evaluation on a unified base model. As the prior work lacks open-source code and weights, the paper does not provide the performance of them on Qwen3-4B. For a fair comparison, we evaluated these algorithms on Qwen2.5-7B-instruct using their reported results and available open-source parameters on LoCoMo:

*Table 8.* Evaluation on the LoCoMo benchmark under a unified `Qwen2.5-7B-Instruct` backbone. This setup decouples algorithmic efficacy from the performance variations of disparate base models.

| Method | F1 | B1 |
|---|---|---|
| Vanilla Baseline | 26.29 | 21.66 |
| LightMem | 37.95 | 33.47 |
| MEM1 | 25.68 | 20.50 |
| MemAgent | 31.97 | 25.63 |
| Memory-R1 | 43.14 | 36.44 |
| **Mem-T** | **55.91** | **50.14** |

The standardized results compiled in Table 8 demonstrate that even when anchored to the exact same neural architecture, `Mem-T` consistently yields a massive performance margin over all alternative paradigms. Specifically, `Mem-T` maintains a $+12.77\%$ F1 and $+13.70\%$ B1 superiority over the strongest competitive memory system (*Memory-R1*). It validates that the observed enhancements are squarely attributable to our hindsight credit assignment and tree-based RL, rather than external advancements in language scaling.

## D. Prompts of `Mem-T`

### D.1. Memory Formation

```
CreateFactTool

class CreateFactTool(BaseTool):
    def __init__(self):
        super().__init__(
            name="create_fact",
            description=(
                "Extract 'Factual Memory' (Concrete, verifiable statements about
                    WHAT happened).\n"
                "CRITICAL RULES:\n"
                "1. Full Entity Scan: Extract attributes and relationships between
                    all entities mentioned (e.g., specific objects, places, third
                    parties), not just the user.\n"
                "2. Pay special attention to time information, including relative
```

```
                        times like 'yesterday' or 'the week before' in the text.\n"
                    "Target two specific types of facts:\n"
                    "1. User Factual Memory: Verifiable facts about the user's events
                        experienced, identity, preferences, items owned, and specific
                        constraints.\n"
                    "2. Environment Factual Memory: Explicit states of the external
                        world, object properties, document knowledge, or tool states, and
                        other entities.\n"
            ),
            parameters={
                "fact": {
                    "type": "string",
                    "description": "The concise, standalone declarative statement. E
                        .g., 'The user prefers Python for backend tasks' or 'The API
                        endpoint v2 is deprecated'."
                },
                "start_time": {
                    "type": "string",
                    "description": "The time when the event occurred or the time
                        when the attribute is valid. Use an empty string if it does
                        not exist."
                },
                "end_time": {
                    "type": "string",
                    "description": "The end time of the event or the expiration time
                        of the attribute. Use an empty string if it does not exist."
                }
            },
            required=["fact", "start_time", "end_time"]
        )
```

### CreateExperienceTool

```
class CreateExperienceTool(BaseTool):
    def __init__(self):
        super().__init__(
            name="create_experience",
            description="Extract 'Experiential Memory' (Actionable lessons, patterns
                , or HOW-TO perform a task)\n"
                "This tool captures lessons learned, reasoning patterns, and
                    executable skills.\n"
                "1. Strategy-based: Reusable heuristics, workflows, or insights
                    derived from reasoning (e.g., 'To solve X, method Y is most
                    efficient').\n"
                "2. Case-based: Key trajectories of Success or Failure that serve as
                     examples (e.g., 'Attempting action A under condition B caused
                    error C').\n"
                "3. Skill-based: Validated code snippets, tool usage protocols, or
                    functions that the agent can execute.\n",
            parameters={
                "experience": {
                    "type": "string",
                    "description": "The distilled content of the experience. It
                        should be formulated as a rule, a cause-effect relationship,
                        or a guideline for future actions."
                },
                "start_time": {
                    "type": "string",
                    "description": "The time when the event occurred or the time
                        when the attribute is valid. Use an empty string if it does
                        not exist."
```

```
                },
                "end_time": {
                    "type": "string",
                    "description": "The end time of the event or the expiration time
                        of the attribute. Use an empty string if it does not exist."
                    }
                },
                required=["experience","start_time","end_time"]
            )
```

---

**UpdatePersonaTool**

```
class UpdatePersonaTool(BaseTool):
    def __init__(self):
        super().__init__(
            name="update_persona",
            description="If there is new and important information about the person,
                such as hobbies, participated projects or significant events, update
                (add or modify) the full character profile for that person.",
            parameters={
                "name": {
                    "type": "string",
                    "description": "Name of the person. Or 'User' if the user does
                        not have a specified name."
                    },
                "profile": {
                    "type": "string",
                    "description": "The full, concise and updated persona text."
                    }
                },
                required=["name", "profile"]
            )
```

---

**UpdateSummaryTool**

```
class UpdateSummaryTool(BaseTool):
    def __init__(self):
        super().__init__(
            name="update_summary",
            description="If there is new information based on the current
                conversation, update the runtime summary of the sessions.",
            parameters={
                "content": {
                    "type": "string",
                    "description": "The concise, complete and updated summary text."
                    }
                },
                required=["content"]
            )
```

## D.2. Memory Evolution

**AddItemTool**

```python
class AddItemTool(BaseTool):
    def __init__(self, vector_db: VectorDBBase, collection_name: str):
        super().__init__(
            name="add_item",
            description="Add a new memory item.",
            parameters={
                "document": {"type": "string", "description": "The content."},
                "turn_time": {"type": "string", "description": "The time of the turn
                    that generated this item."},
                "start_time": {"type": "string", "description": "Start time."},
                "end_time": {"type": "string", "description": "End time."},
            },
            required=["document"]
        )
        self.db = vector_db
        self.collection_name = collection_name
```

**UpdateItemTool**

```python
class UpdateItemTool(BaseTool):
    def __init__(self, vector_db: VectorDBBase, collection_name: str):
        super().__init__(
            name="update_item",
            description="Update an existing memory item.",
            parameters={
                "id": {"type": "string", "description": "The ID of the item to
                    update."},
                "document": {"type": "string", "description": "Enrich the content
                    with more details and update the statistical data or factual
                    frequencies mentioned. Must save the original time information of
                     previously items in the document."},
                "turn_time": {"type": "string", "description": "The time of the turn
                    that generated this update."},
                "start_time": {"type": "string", "description": "New start time."},
                "end_time": {"type": "string", "description": "New end time."},
            },
            required=["id", "document"]
        )
        self.db = vector_db
        self.collection_name = collection_name
```

**DeleteItemTool**

```python
class DeleteItemTool(BaseTool):
    def __init__(self, vector_db: VectorDBBase, collection_name: str):
        super().__init__(
            name="delete_item",
            description="Delete an existing memory item. Use when an item is
                explicitly negated or wrong.",
            parameters={
                "id": {"type": "string", "description": "The ID to delete."}
            },
            required=["id"]
        )
        self.db = vector_db
```

```
            self.collection_name = collection_name
```

**IgnoreItemTool**

```
class IgnoreItemTool(BaseTool):
    def __init__(self):
        super().__init__(
            name="ignore_item",
            description="Do nothing. If the item is completely redundant in both *
                semantic meaning* and *time range*.",
            parameters={
                "reason": {"type": "string", "description": "Reason for ignoring."}
            },
            required=["reason"]
        )
```

## D.3. Memory Retrieval

**SearchSummaryTool**

```
class SearchSummaryTool(BaseTool):
    def __init__(self, vector_db: VectorDBBase):
        super().__init__(
            name="search_summary",
            description="Retrieve relevant summaries to quickly understand the
                context background.",
            parameters={"query": {"type": "string", "description": "Query string
                ."}},
            required=["query"]
        )
        self.db = vector_db
```

**SearchFactsTool**

```
class SearchFactsTool(BaseTool):
    def __init__(self, vector_db: VectorDBBase, top_k: int):
        super().__init__(
            name="search_facts",
            description = "Retrieve 'Factual Memory' (Concrete, verifiable
                statements about WHAT happened).\n"
                "Target two specific types of facts:\n"
                "1. User Factual Memory: Verifiable facts about the user's identity,
                     stable preferences, important events, habits, "
                "historical commitments, and specific constraints.\n"
                "2. Environment Factual Memory: Explicit states of the external
                    world, object properties, "
                "document knowledge, or tool states.\n",
            parameters = {"query": {"type": "string",
                                    "description": "A self-contained, semantically
                                        rich search query rewritten from the user's
                                        intent.\n"
                                            "Instead of raw questions like '
                                                Does he like it?', use specific
                                                declarative queries like "
                                            "'User preference regarding spicy
                                                food' or 'Attributes of Object
                                                X'."}},
```

```
                required=["query"]
            )
            self.db = vector_db
            self.top_k = top_k
```

## SearchExperiencesTool

```
class SearchExperiencesTool(BaseTool):
    def __init__(self, vector_db: VectorDBBase, top_k: int):
        super().__init__(
            name="search_experiences",
            description=(
                "Extract 'Experiential Memory' (Actionable lessons, patterns, or HOW
                    -TO perform a task)\n"
                "This tool captures lessons learned, reasoning patterns, and
                    executable skills.:\n"
                "1. Strategy-based: Reusable heuristics, workflows, or insights
                    derived from reasoning (e.g., 'To solve X, method Y is most
                    efficient').\n"
                "2. Case-based: Key trajectories of Success or Failure that serve as
                     examples (e.g., 'Attempting action A under condition B caused
                    error C').\n"
                "3. Skill-based: Validated code snippets, tool usage protocols, or
                    functions that the agent can execute.\n"
                "Avoid recording raw dialogue history; focus on the distilled '
                    Lesson' or 'Rule'."
            ),
            parameters={
                "query": {
                    "type": "string",
                    "description": (
                        "A self-contained, semantically rich search query rewritten
                            from the user's intent. \n"
                        "Formulate problem-solving queries like 'Standard workflow
                            for analyzing finance reports' "
                        "or 'How to handle TimeoutError in API calls'."
                    )
                }
            },
            required=["query"]
        )
        self.db = vector_db
        self.top_k = top_k
```

## SearchPersonasTool

```
class SearchPersonasTool(BaseTool):
    def __init__(self, vector_db: VectorDBBase):
        super().__init__(
            name="search_personas",
            description="Retrieve character profiles or insights for specific
                individuals.",
            parameters={
                "name": {"type": "string", "description": "Name of the target
                    individual for exact lookup."},
                "query": {"type": "string", "description": "Query string to find
                    personas by traits; ignored if 'name' is provided."}
            },
            required=["query"]
```

```
        )
        self.db = vector_db
```

## SearchTurnsTool

```
class SearchTurnsTool(BaseTool):
    def __init__(self, vector_db: VectorDBBase, top_k: int):
        super().__init__(
            name="search_turns",
            description="Retrieve specific raw dialogue history (Raw Turns). \n"
                        "Use this tool for questions about specific past
                            conversations, verifying exact quotes, or checking 'what
                            was' in detail. \n"
                        "Raw turns provide the most authentic context that summaries
                             or facts might miss.",
            parameters={
                "query": {"type": "string", "description": "Keywords or specific
                    quotes."},
                "top_k": {"type": "integer", "description": "The number of turns to
                    retrieve. Default is 5."}
                },
            required=["query"]
        )
        self.db = vector_db
```

## FinishTool

```
class FinishTool(BaseTool):
    def __init__(self, benchmark_name: str = "locomo", category: str = ""):
        self.benchmark_name = benchmark_name
        self.category = str(category) if category else ""

        description = "Call this when you are confident that you can give the
            correct final answer. Or you should continue to retrieve more information
            ."

        super().__init__(
            name="finish",
            description=description,
            parameters={"answer": {"type": "string", "description": "The concise
                answer following the Final Result Format."}},
            required=["answer"]
        )
```

