# OpenReview forum: "Mem-T: Densifying Rewards for Long-Horizon Memory Agents"
_ICML.cc/2026/Conference — ICML 2026 regular_

### Official Review · Reviewer_MQW1 · 2026-03-10

**Soundness:** 3
**Presentation:** 2
**Significance:** 3
**Originality:** 2
**Overall Recommendation:** 5
**Confidence:** 3

**Summary:**

The paper focus on the memory problem for agents. The authors propose a hierarchical memory system Mem-T, and a corresponding training framework MoT-GRPO. The authors argue for a fully trainable memory framework and overcomes the issue of sparse terminal rewards. Evaluations are performed over long context benchmarks and compared against a range of training-free and training involved methods. Empirical evidence suggests that the proposed method improve both performance and efficiency.

**Compliance With Llm Reviewing Policy:**

Affirmed.

**Key Questions For Authors:**

1. Could the authors comment on how stable the training is?

**Limitations:**

The authors didn't discuss limitation and argues there is no negative societal impact.

**Strengths And Weaknesses:**

Strengths:
The problem is importance and well motivated.
The proposed method appears technically sound. Empirical evidences seems strong, showing consistent improvements across benchmarks.

Weakness:
This paper doesn't provide theoretical guarantees or insights.

---

> ### Author Rebuttal · Authors · 2026-03-31
>
> We appreciate the reviewer's time and effort in reviewing our manuscript!
>
> >**W1: This paper doesn't provide theoretical guarantees or insights.**
>
> We provide two formal theoretical results, which we will incorporate into the revised manuscript.
>
> **1. Proving MoT-GRPO's Lower Gradient Variance for Memory Retrieval Training**
>
> Consider an intermediate node $v_t = \langle a_t, h_t \rangle$ with score function $s_t = \nabla_\theta \log \pi_\theta(a_t | h_t)$. The per-step policy gradient estimators for GRPO and MoT-GRPO are:
>
> $$g_t^{GRPO} = s_t \cdot \hat{A}^{GRPO}_t, g_t^{MoT} = s_t \cdot \hat{A}^{MoT}_t$$
>
> Assume each branch is independently sampled from $h_t$, terminal rewards $R(\tau)$ are conditionally i.i.d. with $\text{Var}[R(\tau) | h_t, a_t] = \sigma^2$.
>
> **GRPO conditional variance:** $\text{Var}[R^{GRPO}_t | h_t, a_t] = \sigma^2$
>
> **MoT-GRPO node reward variance:** Expanding $R(v_t) = \alpha \cdot \text{Evid}(v_t) + \text{Perform}(v_t)$:
>
> $$\text{Var}[R(v_t) | h_t, a_t] = \alpha^2 \cdot \underbrace{\text{Var}[\text{Evid}(v_t) | h_t, a_t]}_{=0 \text{ (deterministic given } a_t)} + \text{Var}[\text{Perform}(v_t) | h_t, a_t]$$
>
> For the performance term, by the variance formula for weighted sums of independent leaf rewards:
>
> $$\text{Var}[\text{Perform}(v_t) | h_t, a_t] = \sigma^2 \cdot \sum_{v_L} w_{v_L}^2$$
>
> By Jensen's inequality, when $|\mathcal{L}(v_t)| \geq 2$ and all $w_{v_L} > 0$:
>
> $$\sum_{v_L} w_{v_L}^2 < \left(\sum_{v_L} w_{v_L}\right)^2 = 1$$
>
> Therefore:
>
> $$\text{Var}[R(v_t) | h_t, a_t] < \sigma^2 = \text{Var}[R^{GRPO}_t | h_t, a_t]$$
>
> Propagating to gradient variance (since $s_t$ is deterministic given $(h_t, a_t)$):
>
> $$\text{Var}[g_t^{MoT}] < \text{Var}[g_t^{GRPO}]$$
>
> The evidence-hit term acts as a zero-variance anchor, and the weighted averaging of child performance further reduces variance relative to any single leaf outcome. Together, these two mechanisms give MOT-GRPO a provably **lower-variance gradient estimator for retrieval training.**
>
> **2. How Densified Rewards Improve SNR for Memory Construction Training**
>
> We formalize the benefit of hindsight credit assignment using the gradient Signal-to-Noise Ratio (SNR):
>
> $$\text{SNR}(\hat{g}) = \frac{\|\mathbb{E}[\hat{g}]\|_2}{\sqrt{\text{Tr}(\text{Var}(\hat{g}))}}$$
>
> Over a long context with $N$ construction operations, only a causal subset of size $k \ll N$ is relevant to any given downstream answer. Under a sparse terminal reward broadcast uniformly to all $N$ operations:
>
> $$\text{SNR}\_{sparse} = \frac{\mathcal{O}(k)}{\sqrt{\mathcal{O}(N \cdot \text{Var}(R_{\text{term}}))}} = \mathcal{O}\left(\frac{k}{\sqrt{N}}\right) \xrightarrow{N \to \infty} 0$$
>
> The gradient becomes pure noise as context length grows.
>
> Our Hindsight Credit Assignment (HCA) densifies rewards via the Retrieval Trace Gate, applying the indicator mask $\mathbb{I}(m \in \mathcal{M}_{v_L})$. For the $N - k$ irrelevant operations, the hindsight score $S(a_i) = 0$ exactly. The gradient then involves only the $k$ relevant actions:
>
> $$\hat{g}\_{dense} = \frac{1}{k} \sum_{i \in \mathcal{K}} \nabla_\phi \log \pi_\phi(a_i) S(a_i)$$
>
> Since the noise variance is now bounded by $k$ (not $N$):
>
> $$\text{SNR}\_{dense} = \frac{\mathcal{O}(k)}{\sqrt{\mathcal{O}(k \cdot \text{Var}(S))}} = \mathcal{O}(\sqrt{k})$$
>
> This is $\mathcal{O}(1)$ with respect to $N$. **HCA truncates the covariance trace from $\mathcal{O}(N)$ to $\mathcal{O}(k)$**, decoupling the SNR from the horizon length and guaranteeing robust credit assignment regardless of context length.
>
> ---
> >**Q1: Could the authors comment on how stable the training is?**
>
> We assess stability across multiple training and inference seeds:
>
> |Training seed|Inference seed|F1|B1|
> |-|-|-|-|
> |1|1|57.90|52.24|
> |1|10|57.41|51.59|
> |10|1|58.45|52.38|
> |10|10|58.73|53.06|
> |42|42|58.65|52.63|
>
> The standard deviation across runs is less than 0.6, indicating stable training.
>
> We additionally monitor the PPO clip ratio during training: following the GRPO official implementation (arXiv:2402.03300), we set $\epsilon = 0.2$; empirically, the clip ratio stays within 0–0.003, indicating that policy updates remain conservative and training does not collapse(arXiv:2508.04349). We will add these stability metrics to the revised paper.
>
> ---
> >**Limitations**
>
> We will discuss additional limitations of our method in the revision, including:
> + Limited base model scale: All experiments use relatively small models (Qwen3-4B and 8B). We will conduct further experiments on larger models.
> + Reliance on gold evidence labels: Although our method can achieve great results without evidence annotations, the strongest performance relies on gold evidence labels. A potential direction is to develop automatic evidence annotation algorithms.
> + Static evaluation setting: Our current evaluation is primarily conducted on static datasets. In the future, we aim to test our method in real-world dynamic production environments.

---

> > ### Author Rebuttal · Reviewer_MQW1 · 2026-04-03
> >
> > Thank the authors for their rebuttal.  For training stability, instead of final metric, it would be worth while to provide learning curve, grad norm etc.

---

> > > ### Author Response · Authors · 2026-04-03
> > >
> > > We sincerely appreciate your insightful feedback and active engagement in the rebuttal discussion!
> > >
> > > We apologize for the earlier misunderstanding. As requested, we have now supplemented our response with learning curves and gradient norm plots across three independent runs (with different random seeds and data splits) for both training stages.
> > >
> > > **Analysis of Memory Retrieval Training Stability:**
> > > As shown in the reward curves(https://anonymous.4open.science/r/MemT_Rebuttal-734C/retrieval_reward.png), all three runs show a consistent reward increase from ~0.3 to ~0.5 by step 200. Despite the expected per-step variance typical of RL, the smoothed trajectories align closely across seeds. Furthermore, gradient norms(https://anonymous.4open.science/r/MemT_Rebuttal-734C/retrieval_grad_norm.png) smoothly decay from ~5.0 to ~2.2 without divergent spikes, confirming that MOT-GRPO provides robust and stable optimization.
> > >
> > > **Analysis of Memory Construction Training Stability:**
> > > The construction training loss curves(https://anonymous.4open.science/r/MemT_Rebuttal-734C/construction_loss.png) demonstrate highly reproducible behavior across seeds, descending rapidly and converging to ~0.22. The gradient norm curves of memory construction training(https://anonymous.4open.science/r/MemT_Rebuttal-734C/construction_grad_norm.png) exhibit a sharp but expected initial peak ( ~16 to ~20) in the first few steps, followed by rapid stabilization to a narrow range of approximately 2.0 to 3.0 for the remainder of training. Notably, an isolated gradient spike around step 155 in Run 2 did not destabilize the loss, highlighting the procedure's resilience.
> > >
> > > Overall, the close agreement across independent seeds confirms smooth convergence and relatively stable training dynamics for both stages.
> > >
> > > While ICML policies prevent us from uploading a revised PDF during this phase, we sincerely commit to incorporating the aforementioned revisions into the final manuscript.
> > >
> > > We sincerely thank you again for your constructive comments and support! Your feedback has significantly contributed to enhancing the quality of our manuscript.

---

### Official Review · Reviewer_dJyk · 2026-03-12

**Soundness:** 3
**Presentation:** 3
**Significance:** 2
**Originality:** 3
**Overall Recommendation:** 4
**Confidence:** 3

**Summary:**

This paper proposes Mem-T, a hierarchical memory agent with four memory stores (working, factual, experiential, and raw memory), together with MoT-GRPO, a training framework intended to address sparse and delayed rewards in long-horizon memory management. The key idea is to convert sparse terminal supervision into denser intermediate supervision by constructing retrieval trees, assigning node-level rewards based on answer quality and evidence retrieval, and then using hindsight credit assignment to supervise memory construction actions. Empirically, the paper reports strong gains on LoCoMo and additional improvements on HotpotQA, LongMemEval, and NarrativeQA, while also claiming better inference-time cost-effectiveness than several memory-agent baselines. Overall, the paper targets an important problem and the empirical gains are notable, but I remain unconvinced that the current evidence fully supports the paper’s strongest claims about generality, fairness, and the source of the gains.

**Compliance With Llm Reviewing Policy:**

Affirmed.

**Key Questions For Authors:**

1. How much of the gain remains if the evidence-based reward is removed or replaced with a supervision-free proxy?

Right now, both retrieval training and construction credit assignment appear to depend heavily on ground-truth evidence annotations. If the method still performs strongly without this privileged signal, my evaluation would improve because it would better support the paper’s claim of addressing sparse-reward training in a generally applicable way. If performance collapses, then the paper should frame itself more narrowly as a method for settings with evidence annotations.

2. Can the authors provide strictly matched baseline comparisons under the same backbone, tool budget, and implementation regime, especially for the strongest trainable baselines?

The current comparison mixes reproduced results, official results, and models of different sizes. If the authors can show that the main conclusions hold under a cleaner apples-to-apples setting, my confidence in the empirical claims would increase substantially. If not, then the reported margins are harder to attribute to the proposed method alone.

3. What is the contribution of each major ingredient beyond the current ablations: tree branching, node-level evidence reward, dual-scale advantage, and hindsight offline construction training?

The current ablations are helpful but still relatively coarse. A more fine-grained decomposition would clarify whether the gains come from the proposed credit-assignment mechanism or from added supervision / search compute. If the improvements remain distributed across the intended components, I would view the methodological contribution as stronger; if one simple ingredient explains most of the gain, the novelty would look more incremental.

4. How stable are the results across random seeds and across different train/validation/test partitions?

This matters especially because the main training benchmark appears small and highly structured. If the authors can show low variance and stable gains, my concern about overfitting and benchmark sensitivity would decrease. If the results vary materially across seeds or splits, then the headline improvements would need to be interpreted more cautiously.

5. What is the full compute cost of training MoT-GRPO relative to simpler alternatives?

The paper emphasizes inference-time efficiency, but the proposed training procedure appears substantially more complex and potentially expensive. If the authors can quantify training cost and show that the gains justify the additional optimization overhead, that would strengthen the practicality argument. Otherwise, the “economical” framing feels incomplete because it only accounts for inference cost.

**Limitations:**

No. The paper includes only a very brief impact statement and does not adequately discuss either methodological limitations or plausible negative societal impacts in a substantive way.

**Strengths And Weaknesses:**

Strengths

1. The paper addresses a real bottleneck in trainable memory agents: sparse terminal rewards over long memory-operation sequences. This is a meaningful and timely problem.

2. The decomposition into memory construction and memory retrieval is intuitive, and the proposed tree-based reward propagation plus hindsight credit assignment is conceptually well aligned with the temporal credit assignment challenge.

3. The reported gains on LoCoMo are substantial, and the paper also includes cross-benchmark evaluation rather than relying on a single dataset.

4. The ablations over memory modules and training components, as well as the sensitivity analysis and qualitative case study, help readers understand which parts of the system matter.

Weaknesses

1. The method relies on privileged supervision more than the paper acknowledges. The dense reward includes an evidence-hit component based on ground-truth evidence, and hindsight credit assignment also uses evidence alignment. This makes the approach less like “pure RL under sparse rewards” and more like a hybrid method with task-specific supervision. That is not inherently invalid, but it weakens the claimed generality.

2. The experimental protocol leaves fairness questions unresolved. Some baselines use different backbone sizes or reported numbers from prior work, while others are reimplemented under Qwen3-4B/Qwen3-8B. This mixture makes the comparisons harder to interpret. The paper does make some effort toward fairness, but the current presentation is still not fully convincing.

3. The source of the improvement is not sufficiently isolated. The paper compares against vanilla GRPO and provides ablations, but it does not adequately disentangle the benefit of tree search, dense evidence reward, dual-scale advantage estimation, hindsight filtering, and the hierarchical memory design itself. As a result, the main novelty claim feels somewhat under-validated.

4. The evaluation is not yet robust enough for the strength of the claims. The training data appears limited, there are no confidence intervals or multi-seed variance reports, and some OOD evaluations are quite small (e.g., the sampled NarrativeQA subset). Given the reported margins and the complexity of the pipeline, stronger statistical and robustness evidence is needed.

---

> ### Author Rebuttal · Authors · 2026-03-31
>
> Thanks for your insightful feedback!
>
> >**W1&Q1: The method relies on privileged supervision**
>
> We agree that the privileged signal is a crucial component, but we emphasize:
> First, **the method works well without the privileged signals**. We provide the ablation on LoCoMo:
>
> |Method|F1|B1|
> |-|-|-|
> |GRPO|53.56|48.33|
> |MoT-GRPO|58.65|52.63|
> |**w/o Evidence**|**56.32**|**50.74**|
>
> Without evidence, our method still surpasses the GRPO. It indicates that dense rewards based on tree structure and the retrieval trace gate provide sufficient signals for training, as it empirically captures which memory actually contributed to successful outcomes.
>
> Second, **how to use the privileged signal is also a contribution**. It provides a principled way to bridge the gap between a delayed, query-level reward and the memory actions that produced the relevant evidence.
>
> ---
> >**W2&Q2: The experiments leave fairness questions unresolved.**
>
> As the prior work lacks open-source code and weights, the paper does not provide the performance of them on Qwen3-4B. For a fair comparison, we evaluated these algorithms on Qwen2.5-7B-instruct using their reported results and available open-source parameters on LoCoMo:
>
> |Method|F1|B1|
> |-|-|-|
> |Baseline|26.29|21.66|
> |LightMem|37.95|33.47|
> |MEM1|25.68|20.50|
> |MemAgent|31.97|25.63|
> |Memory-R1|43.14|36.44|
> |**Ours**|**55.91**|**50.14**|
>
> Mem-T achieved optimal performance on the same backbone, confirming that the gains are attributable to the method rather than backbone differences.
>
> ---
> >**W3&Q3:The source of the improvement is not sufficiently isolated**
>
> We provide additional ablations on LoCoMo:
>
> |Method|F1|B1|
> |-|-|-|
> |Mem-T|58.65|52.63|
> |w/o tree search|55.87|48.58|
> |w/o Evidence|56.32|50.74|
>
> This proves that tree search and the evidence signal are crucial for final performance.
>
> + **Dual-scale advantage:** The individual ablations of $A_{intra}$ and $A_{inter}$ are reported in Table 4 of the paper (1.70 and 4.56 F1 drop).
> + **Hindsight filter:** The ablations of hindsight filter-based memory construction are reported in Table 4 of the paper (3.29 F1 drop).
> + **Hierarchical memory design:** As shown in the 'w/o training' row of Table 2 of the paper, the hierarchical architecture alone achieves 49.38 F1, surpassing nearly all baselines and proving its effectiveness.
>
> ---
> >**W4&Q4:The evaluation is not robust enough**
>
> We trained with multiple seeds and evaluated with multiple inference seeds:
>
> |Training seed|Inference seed|F1|B1|
> |-|-|-|-|
> |1|1|57.90|52.24|
> |1|10|57.41|51.59|
> |10|1|58.45|52.38|
> |10|10|58.73|53.06|
> |42|42|58.65|52.63|
>
> The standard deviation across runs is less than 0.6, and all seeds consistently outperform the GRPO.
>
> We additionally expand evaluation to long-horizon agent benchmarks (GAIA, WebWalkerQA):
>
> |Method|GAIA|WebWalkerQA|LoCoMo|
> |-|-|-|-|
> |Baseline|43.64|54.75|31.50|
> |MemoryBank| — | — | 22.34 |
> |AgentKB|47.27|58.06| — |
> |A-Mem|49.70|57.47|39.43|
> |Memory-R1|—|—|39.25|
> |**Mem-T**|**50.91**|**59.24**|**58.65**|
>
> Methods lacking multi-type memory cannot natively handle the full range of tasks(indicated by "—"). Mem-T achieves the best results across all tasks.
>
> ---
> >**Q5: What's the compute cost of training MoT-GRPO relative to simpler alternatives?**
>
> We provide an analysis comparing MoT-GRPO to GRPO.
>
> **1.Retrieval training token efficiency:**
>
> Let $G$ = number of initial trees, $K$ = Number of branches, $L$ = average trajectory length. Total trajectories generated is $G(1+K)$.
> - GRPO cost: $G(1+K) \cdot L$
> - MoT-GRPO for retrieval: Generating the initial trajectories costs $G \cdot L$. For the new branches, the prefix sequence is reused. So the average length of the branch is $L / 2$ and the branching cost is $(G \cdot K) \cdot L/2$. The total cost is $G \cdot L + G \cdot K \cdot L/2$
>
> Comparing the two: $\text{Ratio} = \frac{1 + K/2}{1 + K} \approx 0.5$
>
> Our method **generates the same number of rollouts as GRPO at half the cost.**
>
> **2.Construction training memory efficiency:**
>
> For Qwen3-4B (fp16, hidden dimension=2560, 36 layers) processing a 40k-token dialogue (~60k construction context), we employ FlashAttention-2 and full gradient checkpointing. The activation cost per rollout is $2\times60k\times2560\times36=10.30GB$.
>
> With a group size of 16, GRPO incurs an activation memory overhead of ~164.79 GB per query.
>
> By shifting to offline memory construction training, we reduce concurrent trajectories from 16 to 1, thereby **cutting activation memory from ~165GB to ~10GB**. And it eliminates the need for a reference model, reducing the overall cost to the SFT level.
>
> **3.Empirical comparison:**
>
> |Method|F1| Cumulative Training Tokens |
> |---|---|---|
> |GRPO|53.56|1.07B|
> |Ours|58.65|0.24B|
>
> MoT-GRPO achieves both higher performance and ~4.5× greater token efficiency.
>
> ---
> >**Limitations**
>
> We will discuss additional limitations of our method in the revision, including its reliance on evidence labels, the limited base model scale, and the static evaluation setting.

---

> > ### Author Rebuttal · Reviewer_dJyk · 2026-04-07
> >
> > Thank you for the detailed rebuttal. I have read the authors’ response carefully, and my main concerns have been adequately addressed. The clarifications and additional evidence strengthen the paper and improve my confidence in the technical soundness of the work. Based on the rebuttal, I maintain my positive assessment of the paper.

---

> > > ### Author Response · Authors · 2026-04-07
> > >
> > > We would like to express our sincere appreciation for your rigorous review and constructive feedback! We are deeply encouraged by your positive appraisal of our work, particularly your remarks that it *addresses a real bottleneck in trainable memory agents* and that *the decomposition into memory construction and memory retrieval is intuitive*.
> > >
> > > We are also pleased that our rebuttal could fully address your concerns regarding *the reliance on privileged supervision*, *the fairness of baseline comparisons*, *the fine-grained module ablations*, and *the training overhead*.
> > >
> > > We are fully committed to integrating all the additional experiments and discussions from the rebuttal into the revised manuscript. **If you feel that these revisions enhance the overall quality of the paper, we would really appreciate it if you might consider reflecting this in your Overall Recommendation.**
> > >
> > > Should you have any further questions or require additional clarification, we are highly receptive to your valuable guidance. Your rigorous scrutiny has been invaluable in refining our research.
> > >
> > > Thank you once again for your dedication, expertise, and encouraging feedback throughout the reviewing process!

---

### Official Review · Reviewer_ay8f · 2026-03-12

**Soundness:** 3
**Presentation:** 3
**Significance:** 2
**Originality:** 2
**Overall Recommendation:** 4
**Confidence:** 4

**Summary:**

Mem-T proposes an LLM-based system that constructs external memory storage based on an incoming text stream (e.g., user-assistant conversation, sequence of documents). Mem-T answers queries by performing iterative retrieval from memory. Mem-T has a set of actions for operating with memory and performing retrieval. The key challenge is sparse supervision and credit assignment over performed actions: the only training signal is final answer quality.

Mem-T addresses this issue with MoT-GRPO, a tree guided RL approach. For retrieval, it builds a tree with agent actions as nodes and propagates training signals from leaf nodes (answers) to intermediate actions. For memory construction, which also requires multiple iterative actions to perform in lack of supervision, Hindsight Credit Assignment is proposed. Memory construction actions are scored using advantages of retrieval actions, gated by overlap with gold evidence texts and whether the produced memory item is retrieved on successful traces. The method was evaluated on LoCoMo, LongMemEval, HotPotQA, and Narrative QA using Qwen3-4/8B models. Mem-T showed improvements over the non-training and GRPO baselines.

**Compliance With Llm Reviewing Policy:**

Affirmed.

**Final Justification:**

Authors addresed several of my key concerns during rebuttal. Also, authors provided plan on revision of text related to presentation and attribution to other methods.

Confidence 3->4
Overall 3->4

Taking into account the promise of a text revision:
presentation: 1->3.

**Key Questions For Authors:**

Consider weaknesses as questions.

- Are there trained baselines using the same Qwen3-4B/8B backbone so results are directly comparable? If not, please add or clarify. Where is the Tree-GRPO baseline?
- Can MoT-GRPO or Tree-GRPO be used to train memory operations online? What is the number of memory operations per sample? At what number of memory operations a tree-like GRPO would be a good choice to use over offline training with Hindsight Credit Assignment, where is a tradeoff in quality/training cost?
- Can Mem-T be effectively trained without gold evidence?
- Why is the memory called "hierarchical" when it mainly consists of four memory types? What is the hierarchy (parent/child structure) or is it just categorical buckets?

**Limitations:**

Limitations are not discussed.

**Strengths And Weaknesses:**

Strengths:
- Addresses an important problem of credit assignment for long context agents: learning what to store and what to retrieve when reward is only answer quality at the end.
- Proposes two distinct ways to densify reward for retrieval and memory actions. 1. Tree based rollouts with bottom-up error propagation MoT-GRPO for retrieval actions. 2. Hindsight Credit Assignment for memory operations, with task and memory specific reward design and offline training.
- Empirical evaluation is relatively broad, including multiple baselines and out-of-domain datasets, plus component ablations.
- The paper also discusses efficiency/token tradeoffs that are important for practical usage.

Weaknesses:
- The novelty and attribution of the Dual-Scale Advantage Estimation and MoT-GRPO is unclear. However, the tree search backup and usage of intra- and inter-tree levels closely match Tree-GRPO (arxiv, 2509.21240). Although MoT-GRPO is mentioned as being "inspired by prior RL methods" (L170-171), the current section on MoT-GRPO reads as if it is a purely novel method. This difference should be explicitly stated by indicating which parts are reused and, ideally, by evaluating it against a Tree-GRPO baseline.
- Used terminology makes the method harder to understand than necessary. The key components are standard tree search, backup, and existing Tree-GRPO training. However, the paper introduces new names, such as Memory Operation Trees (MoT) and Dual-Scale Advantage Estimation. And also posing “MoT-GRPO, a novel RL paradigm”.
- Tree sampling and expansion in Mem-T is mostly standard tree backup / tree search (MCTS-like), not a new method. The novelty is mainly in memory and task specific rewards and credit assignment. The paper should say this clearly. Currently, hiding with naming like  “MoT” makes tree backup look like Mem-T main innovation, and it is confusing to separate what is new vs known under new names.
- “Hindsight Credit Assignment” uses an “Evidence Alignment Gate” that requires gold evidence. The paper claims it can fall back to trace-based credit “in the absence of ground-truth evidence” (L255-257) but it does not report a “no evidence labels” ablation.
- The “Evidence Alignment Gate” ignores what action was performed on the gold evidence, it could be ignore, delete, or incorrect update and still have positive reward, making credit assignment noisy.
- Qwen3-8B results are confusing. Results are in Table 5, but performance being worse or very close to Qwen3-4B is not discussed in the main text and is moved to the appendix. This should be explained and discussed.
- Not clear that trainable baselines are directly comparable as backbone models do not match. Differences may be caused not by the proposed improvements, but from different backbones.


Comments, minor points, typos:
- what does “Crt” in CrtFact/CrtExp/CrtRaw stand for? Consider a clearer prefix or explain it.
- Mem0, A-Mem, MemOS are discussed but do not appear in Table 1.
- L211: clarify what the “reasoning state” z_{k−1}​ is? It was not introduced nor used in other parts of the paper.
- Fig. 5 (Mem-T top-right): possible typo (“20 January”?).
- Table 5 misses vanilla baseline.

---

> ### Author Rebuttal · Authors · 2026-03-31
>
> Thanks immensely for your valuable opinions!
>
> >**W1: Novelty and Attribution Relative to Tree-GRPO**
>
> We agree that Tree-GRPO's tree search and intra/inter-tree advantage are similar to our retrieval training. But the differences are:
> + Tree search: Tree-GRPO builds trees over all operations and trains fully online. We build trees only for short-horizon retrieval; long-horizon construction uses offline hindsight credit assignment.
> + Advantage estimation: Tree-GRPO computes advantages at the trajectory-level and redistributes them to nodes. We compute advantages at the node-level via format correctness, evidence hit rate, and gated credit assignment.
>
> Tree-GRPO faces challenges in memory tasks:
>
> + Scalability: Online tree-based training for >100 construction operations are computationally infeasible (~165GB activation memory for Qwen3-4B, batch size=16, 60k context) and require exponential branching to construct trees.
> + Sparse relevance: A query's reward reflects the quality of only the ~5% relevant memory constructions. Tree-GRPO's layer-wise advantage back-propagation assigns non-zero credit to all nodes, flooding ~95% irrelevant constructions with gradient noise.
>
> **Rather than a trivial extension of Tree-GRPO, we build a memory framework encompassing diverse memory types and operations, and achieve node-level attribution and two-stage training.**
>
> Empirically, we evaluated Tree-GRPO by tuning only the retrieval phase, keeping training costs on LoCoMo acceptable:
> |Method|F1|B1|
> |-|-|-|
> |Tree-GRPO(Retrieval-tuning)|54.20|47.92|
> |Ours(Retrieval-tuning)|55.36|49.51|
> |Ours(Full)|58.65|52.63|
>
> ---
> >**W2: Terminology makes the method harder to understand**
>
> We will clarify in the revision:
> - **MoT-GRPO** is a two-stage training paradigm: online tree-based GRPO for short-horizon retrieval (≤10 steps), offline hindsight credit for long-horizon construction (>100 steps).
> - **Dual-Scale Advantage** refers to intra/inter-tree node-wise advantage, distinct from Tree-GRPO's trajectory-wise computation and redistribution.
> ---
> >**W3: Tree sampling and expansion is not a new method**
>
> We agree that tree sampling is not new, yet it is not our key contribution. Our contributions are:
> 1. A unified memory framework spanning all three functional memory types and all memory operations.
> 2. Joint training of both memory construction and retrieval.
> 3. Dense, attributable signals via tree attribution and hindsight credit assignment.
>
> Tree sampling is only part of Contribution 3. Mem-T overcomes the limitations of previous methods that were optimized for specific memory and operation types without reward attribution.
>
> ---
> >**W4&Q3: Report no evidence labels ablation**
>
> Thanks for your suggestion! The ablation on LoCoMo is as follows:
> |Method|F1|B1|
> |-|-|-|
> |GRPO|53.56|48.33|
> |Tree-GRPO(Retrieval-tuning)|54.20|47.92|
> |MoT-GRPO|58.65|52.63|
> |**w/o Evidence Alignment Gate**|**56.32**|**50.74**|
>
> Without the evidence alignment gate, performance remains above baselines. The gate is valuable but not strictly necessary. And how to utilize evidence is also a contribution.
>
> ---
> >**W5: Evidence Alignment Gate ignores what action was performed on the gold evidence**
>
> The Evidence Gate is an attribution channel, and the advantages assigned to bad behavior can be negative.  Incorrect actions on gold evidence lead to wrong answers, producing negative leaf-node advantages that propagate back as negative hindsight scores, penalizing bad operations.
>
> ---
> >**W6: Qwen3-8B Results Close to 4B**
>
> It is unexpected but externally validated. We kindly clarify that we used the newest Qwen3-4B-Instruct-2507, which community evaluations confirm outperforms Qwen3-8B (mixed thinking) on multiple tasks (see Distil AI's blog: 'We Benchmarked 12 Small Language Models').
>
> ---
> >**W7&Q1: Trainable Baseline Comparability**
>
> Due to character limits, we respectfully refer you to our response to `Reviewer dJyk(W2)`. In comparisons using the identical model, Mem-T is stronger than baselines.
>
> ---
> >**Q2: Can Tree-GRPO Train Memory Online?**
>
> Extremely difficult. As discussed in W1, long operation chains require ~165GB of activation memory and exponentially more branching to construct trees.
>
> Our two-stage design attributes online, short-horizon, and immediate retrieval rewards back to offline construction via the memory pivot. This reduces the construction training overhead to the SFT level.
>
> ---
> >**Q4: Why Called Hierarchical Memory?**
>
> It refers to an abstraction hierarchy: Raw Memory(verbatim turns) → Factual/Experiential Memory(distilled partial turns) → Working Memory(summarizes all turns). Each level derives from lower levels with progressive compression.
>
> ---
> >**Minor Points**
> - Crt = Create memories.
> - Mem0, A-Mem, MemOS omitted for space.
> - $z_{k-1}$: The model-generated reasoning traces and tool calls within $h_{k-1}$.
> - Vanilla Qwen3-8B baseline: F1=33.14, B1=27.86.
>
> ---
> >**Limitations**
>
> We promise to discuss additional limitations in the revision.

---

> > ### Author Rebuttal · Reviewer_ay8f · 2026-04-03
> >
> > I thank authors for their response. Several of my concerns were addressed.
> > My main remaining concern is presentation and attribution: the paper should much more clearly distinguish what is adapted from prior Tree-GRPO / tree-based RL ideas versus what is novel in this work, and avoid presenting standard tree search / backup components as one of key innovations. It should be addresed in text. I also think the important clarifications and additional results from the rebuttal (e.g., Tree-GRPO comparison, no evidence ablation, model details, and limitation) should be incorporated into the paper and not be only present in authors response.

---

> > > ### Author Response · Authors · 2026-04-03
> > >
> > > Thank you very much for your valuable feedback and active engagement in the rebuttal discussion! We fully understand your remaining concern regarding presentation and attribution. To address this, we commit to making the following explicit revisions in accordance with your suggestions:
> > >
> > > * **Section 1, Line 76 (Left):** We will explicitly add: Tree-attributed RL, like Tree-GRPO, is currently used in agent training to densify rewards. However, for long-horizon memory scenarios(Length 40k+, Steps 100+), its online sampling and training overhead is computationally prohibitive, and it fails to resolve the sparse dependency between memory construction and final outcomes.
> > > * **Section 1, Line 105 (Left):** We will clarify our claims by removing "Tree-Guided Optimization" as a primary contribution. Instead, we will highlight: (1) the joint optimization of memory construction and retrieval, and (2) memory-pivoted hindsight credit assignment.
> > > * **Section 2 (Related Work):** We will add a subsection introducing existing tree-based RL (arxiv:2512.08153, arxiv:2509.21240). We will clarify that our algorithm utilizes standard tree-based sampling, and, as detailed in our response to W1, we will explain why these existing methods are ill-suited for long-horizon, sparse-dependency memory domains.
> > > * **Section 3.1, Line 218 (Left):** We will explicitly emphasize: This is the first unified memory framework spanning all three functional memory types (Factual, Experiential, Working) and the full memory lifecycle (Formation, Evolution, Retrieval), making it highly adaptable for long-horizon dialogues and long-horizon agent tasks.
> > > * **Section 3.2, Line 171 (Right):** Rather than merely stating we were "inspired by prior RL methods," we will explicitly specify that our *tree sampling* and *intra/inter-tree advantage computation* during the memory retrieval training phase were inspired by these works.
> > > * **Section 3.2, Line 260 (Left):** We will explicitly contrast our approach with Tree-GRPO: while Tree-GRPO computes and redistributes advantages at the trajectory level, our method performs node-level evaluation based on format and evidence, which better resolves the sparse dependency of final rewards.
> > > * **Section 3.3, Line 224 (Right):** We will explicitly note that algorithms like Tree-GRPO and GRPO suffer from excessive online training overhead and severe gradient noise when facing long-horizon dialogues (40k+) and sparse dependencies (where ~95% of memory actions are outcome-irrelevant), thus necessitating our approach.
> > > * **Section 3.3, Line 273 (Right):** We will explicitly emphasize the contribution: We propose the first joint training of both memory construction and retrieval, whereas previous algorithms typically optimize only a subset of operations.
> > > * **Section 4, Table 2:** We will add the empirical results comparing our method against the Tree-GRPO baseline (as presented in this rebuttal) to clearly demonstrate our performance advantages.
> > > * **Section 4.1, Line 343:** We will add implementation details for the Tree-GRPO baseline. We will explain its exponential tree expansion overhead and out-of-memory issues during the joint training of memory construction and retrieval, and clarify that we ultimately implemented a retrieval-tuning-only version for comparison.
> > > * **Section 5 (Conclusion):** We will clarify that tree sampling is not our novel contribution, but we are the first to apply it to the memory domain. We will emphasize that this is not a trivial migration, but rather driven by three core contributions: (1) the unified memory framework, (2) the joint training of both memory construction and retrieval, and (3) memory-pivoted hindsight credit assignment.
> > >
> > > While ICML policies prevent us from uploading a revised PDF during this phase, we sincerely commit to incorporating all the aforementioned revisions into the final manuscript.
> > >
> > > Thank you again for your expert suggestions and support, which have significantly enhanced the rigor of our manuscript.

---

### Official Review · Reviewer_4kz2 · 2026-03-13

**Soundness:** 3
**Presentation:** 3
**Significance:** 3
**Originality:** 3
**Overall Recommendation:** 4
**Confidence:** 3

**Summary:**

This paper introduces Mem-T, an autonomous memory agent that interfaces with a lightweight hierarchical memory database to perform dynamic updates and multi-turn retrieval. To effectively train long horizon memory management capabilities, the paper proposes MOT-GRPO, which use a retrieval operation tree to obtain step wise supervision.

**Compliance With Llm Reviewing Policy:**

Affirmed.

**Final Justification:**

The author has addressed most of my concerns, and I believe the paper meets the acceptance criteria.

**Key Questions For Authors:**

See weaknesses

**Limitations:**

The authors do not appear to thoroughly discuss the limitations and potential negative societal impacts of their work, and it would be better to explicitly include such a discussion.

**Strengths And Weaknesses:**

Strengths
1. The training method MOT-GRPO is a relatively novel application of tree-based rollouts for memory retrieval.
2. The writing is acceptable, the method description is clear, and the experiments are relatively thorough.

Weaknesses
1. All benchmarks are long-context QA or long-term dialogue QA. It's better to add at least one more type of long-horizon agent tasks. This narrows the scope of claimed generality for “memory agents” to QA-style settings.
2. Computational cost and resource requirements: MOT-GRPO requires building multiple trees and performing multiple rounds of rollouts, although inference tokens is relatively economical, but what is the computational cost threshold for training? How does it differ from standard GRPO? A detailed discussion would be appreciated, as this also relates to the difficulty of reproduction.
3. Although the MoT structure provides some visual visualization and path explanation (as shown in the case study), quantitative analysis of "which nodes/memories are truly crucial" is limited, and a deeper theoretical framework is lacking to characterize the impact of densified reward on credit assignment quality.

---

> ### Author Rebuttal · Authors · 2026-03-31
>
> Thank you for your insightful comments and constructive feedback!
>
> >**W1: Adding Long-Horizon Agent Tasks**
>
> Unlike prior methods that specialize in either experiential memory (suited for long-horizon agents) or factual memory (suited for long-context QA), Mem-T's joint training across all three memory regimes enables seamless generalization to both tasks.
>
> **Experimental Setup:** We conducted experiments on GAIA and WebWalkerQA. Following AgentKB, we leverage Smolagent as the agent framework and GPT-4o for agentic execution. For evaluation, we report Pass@1 on GAIA, and the accuracy on WebWalkerQA under a max 15-step limit.
>
> **Mem-T Workflow:** At task planning and across sub-task boundaries, Mem-T autonomously retrieves relevant memories. During execution, if the context exceeds the window limit, the working memory summarizes the current progress and replaces the overflowed context. Upon task completion, Mem-T distills new experiential and factual memories from the execution trajectory.
>
> |Method|GAIA|WebWalkerQA|LoCoMo|
> |-|-|-|-|
> |Baseline|43.64|54.75|31.50|
> |MemoryBank| — | — | 22.34 |
> |AgentKB|47.27|58.06| — |
> |A-Mem|49.70|57.47|39.43|
> |Memory-R1|—|—|39.25|
> |**Mem-T**|**50.91**|**59.24**|**58.65**|
>
> Methods lacking multi-type memory cannot natively handle the full range of tasks(indicated by "—"). Mem-T achieves the best results across all three tasks.
>
> ---
> >**W2: Computational cost and resource requirements**
>
> We provide an analysis comparing MoT-GRPO to GRPO.
>
> **1.Retrieval training token efficiency:**
>
> Let $G$ = number of initial trees, $K$ = Number of branches, $L$ = average trajectory length. Total trajectories generated is $G(1+K)$.
> - GRPO cost: $G(1+K) \cdot L$
> - MoT-GRPO for retrieval: Generating the $G$ initial full trajectories costs $G \cdot L$. For the new branches, the prefix sequence is reused. So the average length of the branch is $L / 2$ and the branching cost is $(G \cdot K) \cdot L/2$. The total cost is $G \cdot L + G \cdot K \cdot L/2$
>
> Comparing the two: $\text{Ratio} = \frac{1 + K/2}{1 + K} \approx 0.5$
>
> Our method **generates the same number of rollouts as GRPO at half the cost.**
>
> **2.Construction training memory efficiency:**
>
> For Qwen3-4B (fp16, hidden dimension=2560, 36 layers) processing a 40k-token dialogue (~60k construction context), we employ FlashAttention-2 and full gradient checkpointing. The activation cost per rollout is $2\times60k\times2560\times36=10.30GB$.
>
> With a group size of 16, GRPO incurs an activation memory overhead of ~164.79 GB per query.
>
> By shifting to offline memory construction training, we reduce concurrent trajectories from 16 to 1, thereby **cutting activation memory from ~165GB to ~10GB**. And it eliminates the need for a reference model, reducing the overall computational cost to SFT level.
>
> **3.Empirical comparison:**
>
> |Method|F1| Cumulative Training Tokens |
> |---|---|---|
> |GRPO|53.56|1.07B|
> |Ours|58.65|0.24B|
>
> MoT-GRPO achieves both higher performance and ~4.5× greater token efficiency.
>
> ---
> >**W3: Quantitative Analysis of Node Importance and Theoretical Impact of Densified Rewards**
>
> **Quantitative node importance analysis:**
>
> We analyzed 5,646 memory construction nodes by examining the assigned hindsight advantages in relation to the mean F1 scores of their associated queries. The scatter plot is at this anonymous link https://anonymous.4open.science/r/MemT_Rebuttal-734C/fig1.png. The results show extremely strong statistical significance: Pearson $r = 0.867$, Spearman $\rho = 0.926$ ($p < 0.0001$)
>
> This near-perfect rank correlation quantitatively confirms that our framework correctly assigns the highest importance scores to the memory operations that most critically drive successful final answers.
>
> **Theoretical impact of densified rewards on credit assignment quality:**
>
> Due to space limitations, we respectfully refer you to our response to `Reviewer MQW1(W1)` for the full proofs.
> First, we prove that the tree structure of MoT-GRPO yields a lower-variance policy gradient estimator compared to GRPO ($Var[g_t^{MoT}] < Var[g_t^{GRPO}]$). This reduction is achieved because our node reward formulation leverages a deterministic evidence-hit anchor and reduces variance by averaging child performance.
> Second, we demonstrate that our Hindsight Credit Assignment(HCA) effectively decouples the gradient Signal-to-Noise Ratio(SNR) from the context length N. While sparse terminal rewards cause the gradient to degrade into pure noise ($SNR_{sparse} = \mathcal{O}(k/\sqrt{N})$), HCA utilizes an indicator mask to densify rewards. This bounds the noise variance by the k relevant causal operations ($SNR_{dense} = \mathcal{O}(\sqrt{k})$), thereby guaranteeing robust memory construction training over long contexts.
>
> ---
> >**Limitations**
>
> We will discuss additional limitations of our method in the revision, including its reliance on evidence labels, the limited base model scale, and the static evaluation setting.

---

> > ### Author Rebuttal · Reviewer_4kz2 · 2026-04-04
> >
> > Thanks for the response. All my concerns have been addressed. I encourage the authors to add these new experiments to the final version of the paper, and I will maintain my positive score.

---

> > > ### Author Response · Authors · 2026-04-07
> > >
> > > Thank you very much for your insightful review and thoughtful follow-up! We are truly grateful for your positive feedback, especially the kind comments regarding *the novelty of our method*, *the clarity of the writing*, and *the thoroughness of the experiments*.
> > >
> > > We are also glad to hear that our rebuttal fully resolves your concerns regarding *the generalization to broader long-horizon agent tasks*, *the comparisons of training computational costs*, and *the quantitative analysis of node importance*.
> > >
> > > As suggested, we will ensure that these new experiments and discussions from the rebuttal are incorporated into the final manuscript. **If you feel that these additions strengthen the paper’s overall quality, we would be very grateful if you might consider reflecting that in the Overall Recommendation.**
> > >
> > > If you require further clarification on any points, we remain entirely at your service. We deeply value your thoughtful feedback, as it has helped us improve the quality of our paper.
> > >
> > > Thank you again for your careful reading, constructive feedback, and expertise!

---

### Decision · Program_Chairs · 2026-04-30

**Decision:**

Accept (regular)

**Comment:**

Reviewers agreed that this paper addresses an important problem for long-horizon memory agents and that the paper provides a technically meaningful attempt to improve credit assignment for both retrieval and memory construction. Several reviewers highlighted the practical importance of the problem setting, the combination of tree-guided retrieval training and hindsight credit assignment, and the breadth of the empirical study, including the reported performance/efficiency tradeoff.

The main concerns centered on contribution attribution and presentation rather than on a fatal soundness flaw. In particular, one reviewer remained concerned that the tree-based retrieval component is close to prior tree-based RL methods and asked for clearer attribution and positioning, while others asked for clearer discussion of training cost, gold-evidence dependence, and the interpretation of the densified rewards. The rebuttal materially improved the paper: two reviewers stated that their concerns were fully resolved, one reviewer increased the score from reject to weak accept, and the remaining reservations were mainly about novelty framing and clearer discussion of limitations. After considering the reviews, rebuttal, and discussion, my assessment is that the paper is technically sound and useful, but that its contribution would be better positioned as a solid step forward rather than a clear medium-priority accept. I therefore recommend weak accept.